# Genetic and functional diversification of chemosensory pathway receptors in mosquito-borne filarial nematodes

**Nicolas J. Wheeler**, **Zachary W. Heimark**, **Paul M. Airs**, **Alexis Mann**, **Lyric C. Bartholomay**, **Mostafa Zamanian**\*

Department of Pathobiological Sciences, University of Wisconsin-Madison, Madison, Wisconsin, United States of America

\* mzamanian@wisc.edu

**Data Availability Statement:** All raw data and scripts used for comparative genomics, phylogenetics, data analysis, and data visualization are publicly available at https://github.com/

## Abstract

Lymphatic filariasis (LF) afflicts over 60 million people worldwide and leads to severe pathological outcomes in chronic cases. The nematode parasites (Nematoda: Filarioidea) that cause LF require both arthropod (mosquito) intermediate hosts and mammalian definitive hosts for their propagation. The invasion and migration of filarial worms through host tissues are complex and critical to survival, yet little is known about the receptors and signaling pathways that mediate directed migration in these medically important species. In order to better understand the role of chemosensory signaling in filarial worm taxis, we employ comparative genomics, transcriptomics, reverse genetics, and chemical approaches to identify putative chemosensory receptor proteins and perturb chemotaxis phenotypes in filarial worms. We find that chemoreceptor family size is correlated with the presence of environmental (extra-host) stages in nematode life cycles, and that filarial worms contain compact and highly diverged chemoreceptor complements and lineage-specific ion channels that are predicted to operate downstream of chemoreceptor activation. In *Brugia malayi*, an etiological agent of LF, chemoreceptor expression patterns correspond to distinct parasite migration events across the life cycle. To interrogate the role of chemosensation in the migration of larval worms, arthropod and mammalian infectious stage *Brugia* parasites were incubated in nicotinamide, an agonist of the nematode transient receptor potential (TRP) channel OSM-9. Exposure of microfilariae to nicotinamide alters intramosquito migration, and exposure of L3s reduces chemotaxis toward host-associated cues in vitro. Nicotinamide also potently modulates thermosensory responses in L3s, suggesting a polymodal sensory role for *Brugia osm-9*. Reverse genetic studies implicate both *Brugia osm-9* and the cyclic nucleotide–gated (CNG) channel subunit *tax-4* in larval chemotaxis toward host serum, and these ion channel subunits partially rescue sensory defects in *Caenorhabditis elegans osm-9* and *tax-4* knock-out strains. Together, these data reveal genetic and functional diversification of chemosensory signaling proteins in filarial worms and encourage a more thorough investigation of clade- and parasite-specific facets of nematode sensory receptor biology.

zamanianlab/BrugiaChemo-ms. The optical flow algorithm for motility analysis is available at https://github.com/zamanianlab/BrugiaMotilityAnalysis. Short-read and long-read sequencing data has been deposited into NIH BioProjects PRJNA548881 and PRJNA548902, respectively. An interactive version of Fig 1 and S2 Fig is available at https://zamanianlab.shinyapps.io/ChemoR/, where chemoreceptor annotation and amino acid sequence data is available for download.

**Funding:** Funding for MZ is provided by NIH NIAID K22 (K22AI125473, NIH.gov) and R01 (R01AI151171,NIH.gov) grants, the Wisconsin Alumni Research Foundation (WARF, warf.org), and the National Center for Veterinary Parasitology (NCVP, ncvetp.org). Funding for LCB is provided by an NIH NIAID grant (R21AI117204). The funders had no role in study design, data collection and analysis, decision to publish, or preparation of the manuscript.

**Competing interests:** The authors have declared that no competing interests exist.

**Abbreviations:** CI, chemotaxis index; CNG, cyclic nucleotide–gated; DEC, diethylcarbamazine citrate; DPE, days postextraction; DPI, days postinfection; dsRNA, double-stranded RNA; FBS, fetal bovine serum; GPCR, G protein–coupled receptor; HPE, hours postextraction; IVM, ivermectin; L1, first stage larvae; L2, second stage larvae; L3, third stage larvae; L4, fourth stage larvae; LF, lymphatic filariasis; LVP, Liverpool strain; MDA, mass drug administration; mf, microfilaria; NAM, nicotinamide; qPCR, quantitative PCR; RNAi, RNA interference; RNA-seq, RNA sequencing; TRP, transient receptor potential; TPM, transcripts per million.

## Introduction

Lymphatic filariasis (LF) is a parasitic disease caused by mosquito-borne filarial worms (Nematoda: Filarioidea) belonging to the genera *Wuchereria* and *Brugia*. LF is estimated to affect over 60 million people worldwide, particularly in impoverished tropical regions [1]. Infections are associated with chronic disability and physical disfigurement, most commonly resulting from advanced manifestations of lymphedema, hydrocele, and elephantiasis. These conditions yield additional stigmatization and mental health burdens on those suffering, which in turn can prevent individuals from seeking treatment [2–4]. Currently, chemotherapeutic control of LF is mainly achieved through mass drug administration (MDA) of diethylcarbamazine citrate (DEC), ivermectin (IVM), albendazole, or combinations of these anthelmintic drugs [5,6]. However, the suboptimal efficacy of available drugs against adult parasites, contraindication of DEC and IVM in patients with multiple filarial diseases, and threat of drug resistance underlie efforts to develop new treatment options. A better understanding of the molecular basis of parasite behaviors required for successful transmission and parasitism has the potential to aid LF control efforts.

The filarial worms that cause LF have complex life cycles that require migration through hematophagous arthropod intermediate hosts and mammalian definitive hosts [7]. Microfilariae (mf) released from viviparous females in the human lymphatics must reach the peripheral blood, where they can be ingested through the proboscis of feeding mosquito vectors. In susceptible mosquitoes, larvae burrow out of the midgut, pass through the hemocoel, and invade cells of the thoracic flight muscles. Over the course of approximately 2 weeks, larvae grow and develop to the human-infective third stage larvae (L3) and migrate to the mosquito head region in preparation for transmission to the mammalian host [8,9]. L3s are deposited onto the skin of hosts from the proboscis of feeding mosquitoes and must quickly travel through the bite wound and connective tissues to reach the lymphatic system, where they reach sexual maturity [10–12]. Although the life cycle of LF parasites is well described, the molecular basis for stage-specific migratory behaviors is unknown.

There is growing evidence that chemosensation and other sensory modalities play an important role in nematode parasite transmission and intrahost migration [13–22]. However, most studies have focused on single-host nematode parasites with direct life cycles, which are phylogenetically distant from the vector-borne filarial parasites of clade III [23]. Recent studies using human-infective *Brugia malayi* and feline-infective *B. pahangi*, a model species for human LF, reveal the presence of canonical nematode sensory organs (amphids) and robust chemotaxis responses to host-associated cues in vitro [24–27]. Filarial worms also exhibit genus-specific patterns of migration within the same host species [28]. These observations strongly suggest an important role for chemosensation and chemotaxis in LF parasitism and provide motivation to dissect the signaling pathways and mediators of sensory behaviors in these medically important parasites.

Chemosensory signaling pathways in the model nematode *Caenorhabditis elegans* are well characterized [29]. G protein–coupled receptors (GPCRs) function as chemoreceptors at the amphid cilia, and activation leads to signaling through cyclic nucleotide–gated (CNG) channels or transient receptor potential (TRP) channels, depending on cell type [30–33]. Each amphid neuron expresses a diverse array of GPCRs, in contrast to the one-receptor-per-cell model in vertebrates [34–36]. These pathways have likely evolved to reflect the diversity of ancestral nematode life-history traits and environmental cues encountered by different nematode species [14,19–21]. Despite superficial conservation of nematode chemosensory pathways, we hypothesized that there are important differences in repertoire, patterns of

expression, and function of chemosensory genes among free-living, single-host, and vector-borne parasitic nematodes belonging to diverse clades [23,37].

Here, we investigate nematode chemosensory receptor biology in LF parasites and connect in vitro and in vivo chemotaxis behaviors to chemosensory signaling pathways. We carry out genomic and transcriptomic analyses of putative chemosensory GPCRs (chemoreceptors), CNG channels, and TRP channels in a panphylum context. Using a combination of chemical and reverse genetic approaches, we present the first evidence of *Brugia* chemotaxis behaviors modulated by specific sensory-associated receptors. Lastly, we explore how these data reveal unique aspects of chemosensory biology in these medically important parasites.

## Results

### Filarial worms contain a compact and unique repertoire of chemoreceptors

To elucidate the putative chemosensory pathway of mosquito-borne filarial worms and to identify and annotate chemoreceptors, we first performed a panphylum analysis of 39 nematode genomes [38,39], representing all published filarial genomes and high-quality assemblies across four primary nematode clades [23] (S1 Table, S1 Fig). In total, 10,440 putative chemoreceptor genes were identified and confidently classified within superfamilies (Str, Sra, Srg) or "solo" families (srz, sro, srsx, srbc, srxa, sra) (S1 File) [40]. Although the majority of receptors were also annotated at the family level, some clade IIIa/b and clade IV chemoreceptors did not clearly group with the families that were originally described in *C. elegans* (Fig 1B and 1C, S1 Data). However, each of the 30–100 chemoreceptors found in filarial worm species (clade IIIc) was readily classified into the established 23 nematode chemoreceptor families [40,41]. Within these families, we found no one-to-one orthologs between filarial parasites and species belonging to other clades, demonstrating the divergence of the filarial chemoreceptor subset. Instead, there have been clear paralogous gene radiations that have resulted in enrichment of the *srx*, *srab*, *srbc*, and *srsx* families (Fig 1B and 1C). Filarial parasites also contain relatively numerous *srw* receptors, but these likely include neuropeptide receptors in addition to some chemoreceptors of environmental peptides [41,42].

Filarial worm genomes contain a reduced subset of chemoreceptors when compared with other parasitic and free-living nematodes, including *C. elegans* and *C. briggsae* (clade V), both of which contain over 1,200 chemoreceptors (Fig 1B) [36,40,41]. Although it is known that many parasitic nematodes contain fewer chemoreceptor genes compared with *C. elegans* [43,44] and, indeed, often fewer genes in total [45], our panphylum analysis revealed a significant correlation between chemoreceptor gene count and the presence and nature of free-living or environmental stages of each nematode species life cycle (Fig 1D, S2 File). Nematodes that are parasitic and are host contained or lack motile environmental stages exhibit more-compact chemoreceptor repertoires than those that are exclusively free living or contain free-living stages (Spearman's rank-order correlation, $\rho = -0.813$, $p = 3.24 \times 10^{-10}$), and this is not a function of genome contiguity or completeness (S2 Fig). This correlation has been observed when comparing smaller numbers of nematodes, and our comprehensive approach confirms this pattern across the phylum [40,43,44,46].

Chromosomal synteny between *B. malayi* and *C. elegans* further illustrates the divergence of chemoreceptors in the Filarioidea (Fig 2). The majority of *C. elegans* chemoreceptors are found on chromosome V (67%), and chemoreceptor genes likely underwent several birth–death cycles that reflect the adaptive needs of encountering new locales [40]. Putative *B. malayi* chemoreceptors are primarily found on chromosomes II (31%) and IV (35%) and are clustered by family, suggesting lineage-specific gene duplications.

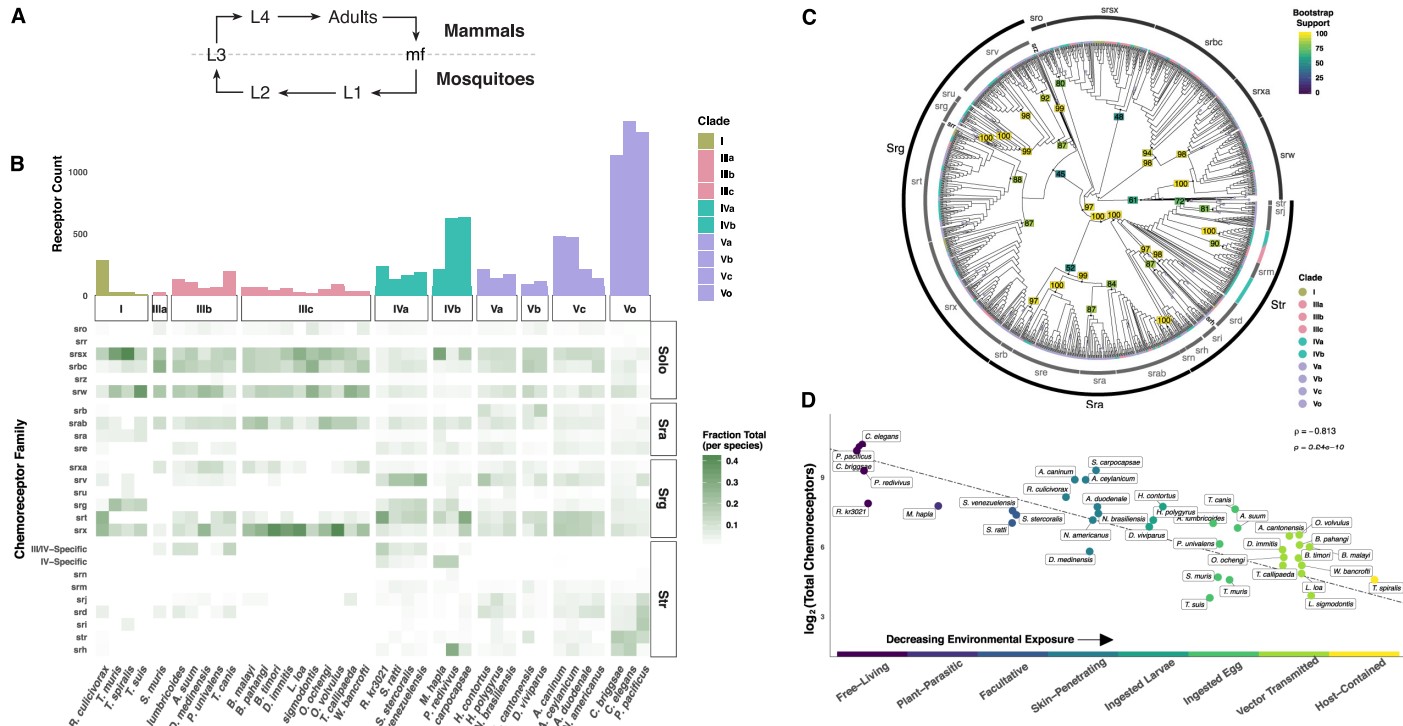

**Fig 1. The genomes of filarial worms contain a reduced complement of divergent chemoreceptors.** Chemoreceptors were mined from 39 nematode genomes, and the phylogeny of chemoreceptors from a down-sampled species set was constructed with ML inference. (A) General life cycle of mosquito-transmitted filarial worms. (B) Filarial worm (clade IIIc) genomes contain far fewer chemoreceptors than other nematodes, and they are enriched for *srsx*, *srab*, *srbc*, and *srx* receptors. Each box in the heatmap is normalized to the total number of chemoreceptors per species. (C) Family and superfamily categorizations from *C. elegans* were used to annotate the final phylogeny. Clade IIIc chemoreceptors are diverged from *C. elegans* and other nematodes, without any one-to-one homologs. Filarial worm chemoreceptors are notably diverged in *srsx*, *srab*, *srbc*, and *srx*. Nodal values represent percent bootstrap support of 1,000 separate replicates. Branches consisting of only *C. elegans* receptors were collapsed to aid visualization. (D) A decrease in chemoreceptor count is correlated with an increase in extrahost (e.g., terrestrial) stages within nematode life cycles. Completely free-living nematodes such as *C. elegans*, *C. briggsae*, *Pristionchus pacificus*, and *Panagrellus redivivus* have many more chemoreceptors than parasitic nematodes that are vector transmitted or host contained such as the filarial worms and *Trichinella spiralis*. Note that the x-axis is categorical, and slight jitter has been added to the points to decrease point/label overlap. ρ was calculated with Spearman's rank correlation with the null hypothesis that ρ = 0. Raw data for (B) and (D) can be found at https://github.com/zamanianlab/BrugiaChemo-ms. Raw tree data for (C) can be found in Newick format in S1 Data. L1, first stage larvae; L2, second stage larvae; L3, third stage larvae; L4 fourth stage larvae; mf, microfilaria; ML, maximum-likelihood.

## *B. malayi* chemoreceptors are associated with sensory tissues and display stage-specific expression patterns

These comparative data indicate that arthropod-borne filarial worms rely on a small complement of clade- and species-specific chemoreceptors to interact with and navigate their host environments. Nematode chemosensation is primarily mediated by anterior amphid sensory structures, and many nematodes also possess caudal chemosensory phasmids associated with male sex organs that likely aid in copulation. In *C. elegans* hermaphrodites and *Strongyloides stercoralis*, a soil-transmitted helminth, chemoreceptors and sensory pathway effectors have been primarily localized to these anterior and posterior structures but can also be found in other nonneuronal cells [13,36]. We examined the expression of chemoreceptors in structures implicated in adult filarial worm chemosensation [24,47,48] using RNA sequencing (RNA-seq) of anterior and posterior tissues. *B. malayi* female head, male head, and male tail tissue regions were excised for RNA-seq detection of anterior, posterior, and sex-specific chemoreceptor transcripts. Most chemoreceptor transcripts are preferentially detected in one of these

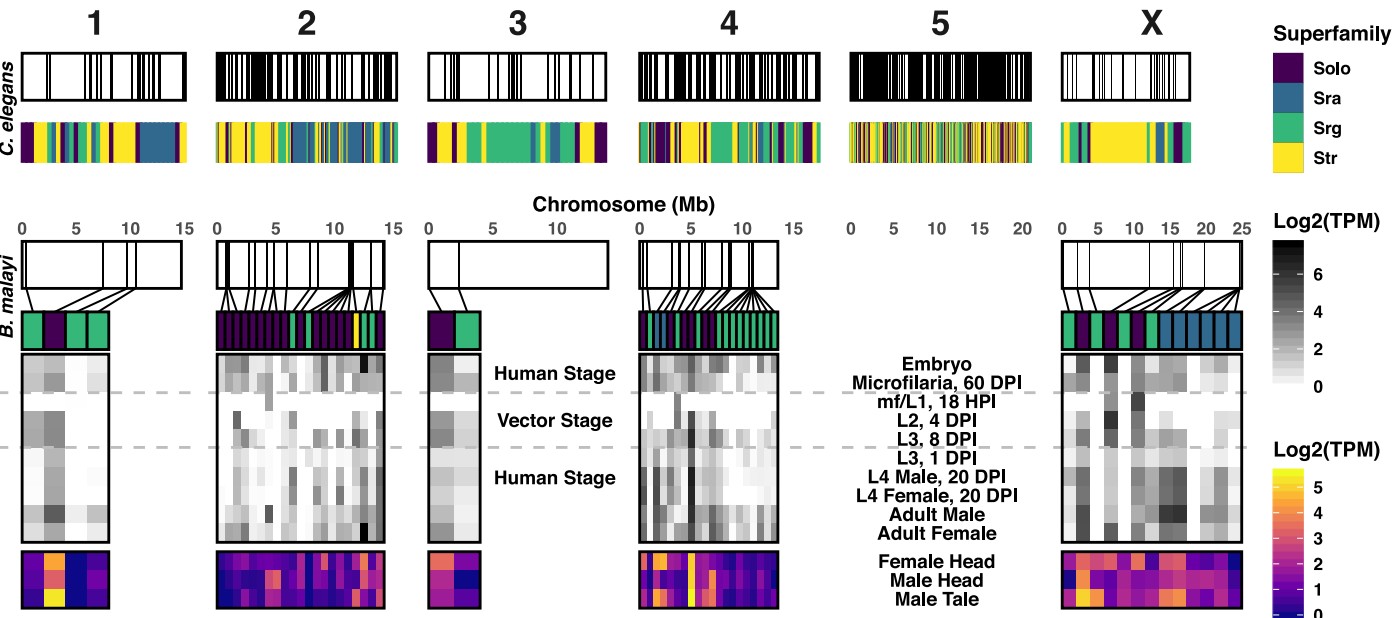

**Fig 2. Chemoreceptors are clustered in the *B. malayi* genome and are enriched in specific life stages and adult tissues.** The chromosomal location of annotated *B. malayi* and *C. elegans* chemoreceptor genes are shown with chromosomes in white, and chemoreceptor loci are depicted as black lines. *C. elegans* chemoreceptors are found throughout the genome but are heavily clustered on chromosome V, and these clusters can be enriched for specific families and superfamilies [40]. Likewise, *B. malayi* chemoreceptors are clustered on chromosomes II and IV. RNA expression data reveal distinct patterns of chemoreceptor expression across the life cycle and in discrete adult male and female tissues. Raw data can be found at https://github.com/zamanianlab/BrugiaChemo-ms. DPI, days postinfection; HPI, hours postinfection; L1, first stage larvae; L2, second stage larvae; L3, third stage larvae; L4, fourth stage larvae; mf, microfilaria; TPM, transcripts per million.

disparate anatomical regions, although a small number show a broader distribution of expression across these regions (Fig 2, S3 Fig).

We further hypothesized that the unique cues encountered by filarial parasites across developmental time points would be reflected by stage-specific chemoreceptor expression patterns. In *C. elegans*, chemosensory processes coordinate movement toward food or mates and away from pathogens, predators, or noxious substances [49–52]. In contrast to the open and less predictable environments navigated by free-living nematodes, filarial worms encounter distinct environmental niches that have strictly patterned transitions. We used staged transcriptomes to analyze the expression of chemoreceptors across the life cycle of *B. malayi* [53] and identified receptors that correspond to migratory landmarks throughout the parasite life cycle (Fig 2). Expression data show that mf circulating in the bloodstream at 60 days postinfection (DPI) express a larger number of chemoreceptors compared with nonmigratory first stage larvae (L1) and second stage larvae (L2) larvae that are contained within mosquito muscle cells; for instance, only four chemoreceptors during the mf/L1 stage in the mosquito have detectable expression (Fig 2). Interestingly, there is also a large number of chemoreceptors expressed in embryos. It is possible that there is transcript buildup in embryos in preparation for release from the adult female. The similarity between embryo and adult female expression patterns also suggests that there may have been some level of contamination during the difficult isolation of embryos from gravid females. There is an increase in chemoreceptor representation and expression during the migratory and mammalian-infective L3, as well as in later mammalian stages that undergo migration and potentially engage in mate-seeking behaviors. Together, these analyses show that *B. malayi* expresses distinct sets of chemoreceptors in a sex-, tissue-, and stage-specific manner.

## Filarial worms have a divergent subset of downstream chemosensory pathway receptors

In *C. elegans*, ligand binding to chemoreceptors activates heterotrimeric G proteins that ultimately produce neuronal depolarization via the opening of CNG or TRP channels, depending upon cell type [29,36]. The CNG channels TAX-4 and TAX-2 mediate signaling in amphid neurons ASE, AWC, AWB, ASI, ASG, ASJ, and ASK, whereas the TRPV (vanilloid-type) channels OSM-9 and OCR-2 are necessary for signaling in AWA, ASH, ADF, and ADL [29]. To assess the conservation of these downstream signaling pathways in filarial parasites, we mined TRP and CNG channels across nematode genomes to examine interspecies variation in ion channel complements.

We found that filarial worms contain one-to-one homologs of *osm-9* but do not have homologs of *ocr-3*, *ocr-4*, *trpa-1*, *pkd-2*, *trp-1*, or *gtl-1* (Fig 3A, 3C and 3E, S2 Data). Filarial parasites contain two *ocr-1/2*–like genes (*Bm5691* and *Bm14098*), but these are more closely related to each other than they are to *C. elegans ocr-1* or *ocr-2* (Fig 3E). In *C. elegans*, OSM-9 and OCR-2 are mutually dependent upon each other for intracellular trafficking to sensory cilia [33]. Cell-specific TRP channel expression patterns and TRP subunit interactions are unknown in filarial parasitic species, and it is not clear which filarial parasite subunit might provide a homologous Cel-OCR-2 ciliary targeting function or, indeed, whether such a trafficking function is necessary. Interestingly, we found that *Bma-ocr-1/2a* (*Bm5691*) is expressed in the female head (transcripts per million [TPM] > 2.5) but is found in very low abundance in the male head and tails (TPM < 1), whereas the opposite is true for *Bma-ocr-1/2b* (*Bm14098*). On the other hand, *Bma-osm-9* is found at a relatively high abundance in both male (TPM > 15) and female (TPM > 18) heads. This tissue distribution of transcripts could indicate the potential for sex-specific subunit interactions among these TRPV channels. Although missing in clade IIIc filarial parasites, we found homologs of *ocr-3*, *ocr-4*, *pkd-2*, and *trp-1* in other clade III species (e.g., soil-transmitted ascarids), and the most parsimonious explanation of their absence in filarial worms is that these genes were lost sometime after the divergence of Spirurida and Ascarida [37]. Conversely, *trpa-1*, which functions in *C. elegans* mechanosensation in QLQ [54], and *gtl-1*, which functions in ion homeostasis in the *C. elegans* intestine [55], appear to be specific to clade V.

Similarly, filarial worms have one-to-one homologs of *tax-4* (α-type) and *tax-2* (β-type) CNG channel subunits but lack *cng-1* and *cng-3* (Fig 3B, 3D and 3F, S3 Data). Filarial worm genomes possess a third CNG (*Bm7148*) that is related to both *cng-2* and *che-6*, but phylogenetic analysis suggests the divergence of *cng-2* and *che-6* to have occurred later than the most recent common ancestor of the Filarioidea and *C. elegans*, making it difficult to ascribe putative function to the *cng-2/che-6* homolog in Filarioidea. In *C. elegans*, TAX-2 and TAX-4 are broadly expressed in amphid sensory neurons and mediate both thermosensory and chemosensory function, whereas other channels, like CNG-2 and CHE-6, have more-restricted expression patterns and modulate these pathways [56]. It is unclear whether *Bma*-TAX-4 and *Bma*-TAX-2 coordinate multiple sensory modalities as in *C. elegans* and whether Bm7148 interacts with these proteins and pathways. *Bm7148* is highly expressed in all three tissues that we analyzed (TPM > 20), whereas *Bma-tax-4* and *Bma-tax-2* have TPM values of less than 2 in all cases.

## Treatment with a nematode TRPV agonist inhibits chemoattraction but not chemoaversion of infective-stage *Brugia* larvae

Our bioinformatic analyses show that filarial worms have evolved divergent sets of chemoreceptors but maintain much of the core structure of the chemosensory pathway as modeled in

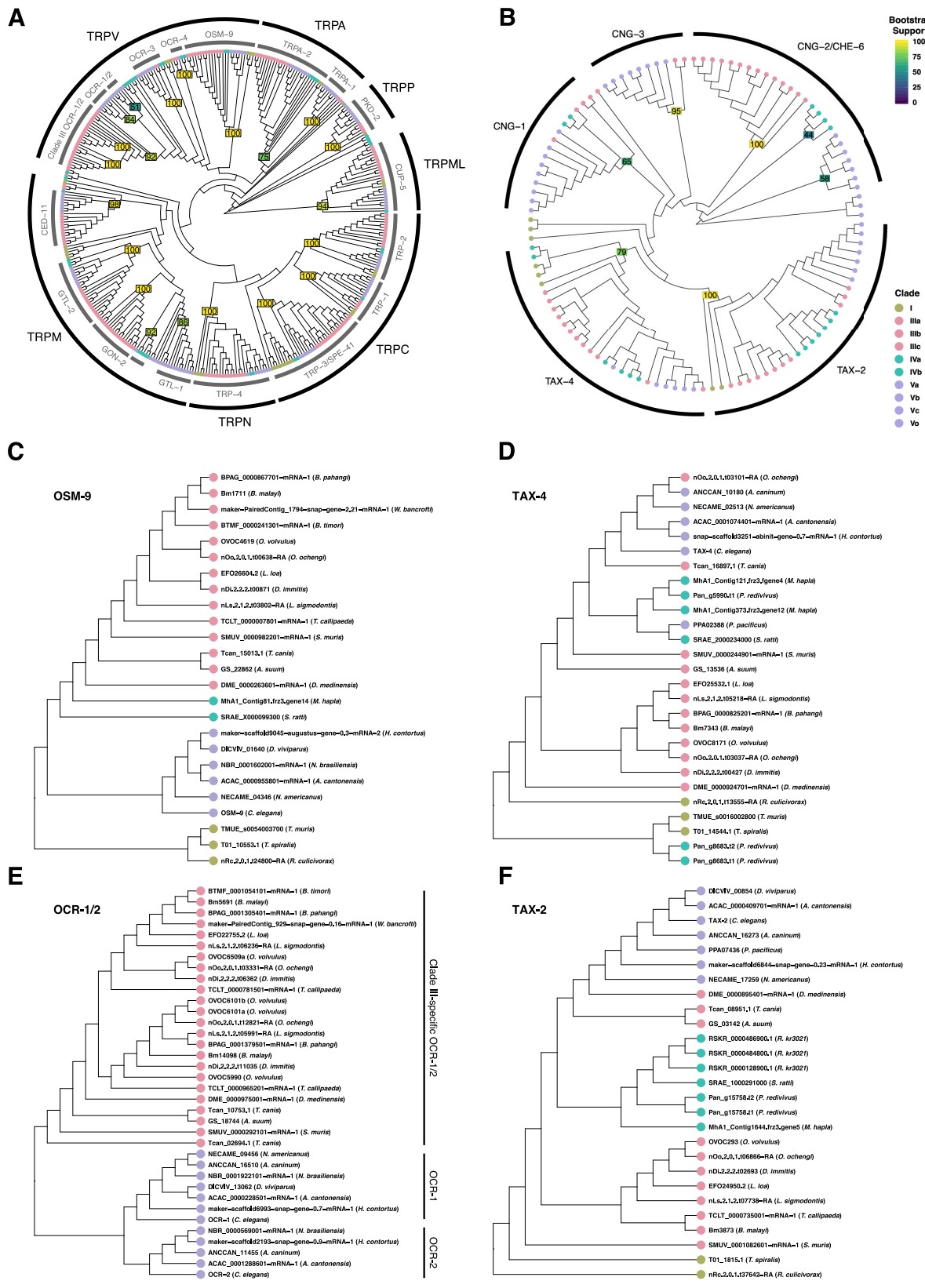

**Fig 3. Filarial worms possess unique complements of broadly conserved nematode TRP and CNG channels.** The phylogenies of (A) TRP and (B) CNG channels were constructed with Bayesian inference. Nodal values represent the posterior probability. (C) *osm-9* and (E) *ocr-1/2* subtrees were drawn from (A), and (D) *tax-4* and (F) *tax-2* subtrees were drawn from (B). Filarial worms have one-to-one orthologs of *C. elegans osm-9*, *tax-4*, and *tax-2*. In contrast, the two *ocr-1/2*–like genes from filarial worms are more closely related to each other than with the homologous *Cel-ocr-1* and *Cel-ocr-2* and belong to a diverged clade IIIc grouping of OCR-1/2–like channel subunits. Raw tree data for (A) and (B) can be found in S2 Data and S3 Data, respectively. CNG, cyclic nucleotide–gated; TRP, transient receptor potential.

*C. elegans*. To test conservation of chemosensory function in TRPV channels of filarial worms, we treated infective-stage *Brugia* L3s with nicotinamide (NAM), an agonist of the *C. elegans* OSM-9/OCR-4 heteromeric channel [57], and measured chemotactic responses to host-associated cues. These experiments were performed with *B. pahangi*, a model *Brugia* species [24,25,27]. *B. pahangi* L3s freshly extracted from infected *Aedes aegypti* Liverpool strain (LVP) mosquitoes are strongly attracted to both fetal bovine serum (FBS) and sodium chloride but are weakly repelled by 3-methyl-1-butanol (a component of human sweat attractive to *S. stercoralis* and *Anopheles gambiae*) (8,13,49) (Fig 4B). Treatment of freshly extracted L3s with 250 μM NAM significantly reduced chemoattraction to serum (44.8% reduction) and sodium chloride (73.1% reduction) but had no significant effect on aversion to 3-methyl-1-butanol (Fig 4B). NAM treatment did not impact worms' overall translational movement on the chemotaxis plates (Fig 4C), indicating that NAM causes a specific defect in chemotaxis rather than a general depression in movement ability.

To ensure that the *Bpa-osm-9*, the putative target of NAM, was expressed during the performance window of our assay and that expression was not altered by the ambient temperatures that L3s experience during assay preparation, we measured the relative expression of *Bpa-osm-9* in L3s immediately after extraction from mosquitoes and after 4 hours of in vitro culture at human body temperature (37°C) or ambient temperature (approximately 21°C). The relative expression of *Bpa-osm-9* was unchanged over this time frame at either temperature ($p$ = 0.4215, Fig 4D).

Although the expression of *Bpa-osm-9* does not change following extraction (Fig 4D), parasites maintained overnight in complete media do not show a chemotactic response to serum, even with preassay incubation in serum-free media, and show a reduced motility on the chemotaxis plate when compared with untreated freshly extracted parasites (Fig 4E). Although it is possible that the specific unknown chemoreceptors involved in serum response are downregulated by this time point, it is more likely that artificial culture conditions have effects on parasite health that compromise chemotactic potential. These results highlight the importance of using freshly extracted L3s in these assays.

## Treatment with a nematode TRPV agonist alters an infective-stage *Brugia* larvae thermosensory response

L3s that have departed the intermediate mosquito host are challenged with stark temperature shifts from the ambient temperature in the mosquito to warmer temperatures on the definitive host's skin of approximately 24–34°C [58] to an even warmer host core temperature of 37°C. During in vitro culture, healthy L3 worms elongate and vigorously thrash in warm medium, but thrashing will transition to coiling and reduced motility as the culture medium cools. The coiling and reduction in motility caused by cooling is reversed after returning the parasites to 37°C (S4 Fig), indicating that the phenotype is not a result of general sickness or tissue damage. In the course of performing L3 chemotaxis experiments with NAM, we noticed that treated L3s had a reduced coiling response. To confirm this effect, we performed dose–response experiments and video-recorded parasites exactly 20 minutes after transfer from 37°C to room temperature, the point at which untreated parasites tightly coil. Both blinded manual scoring

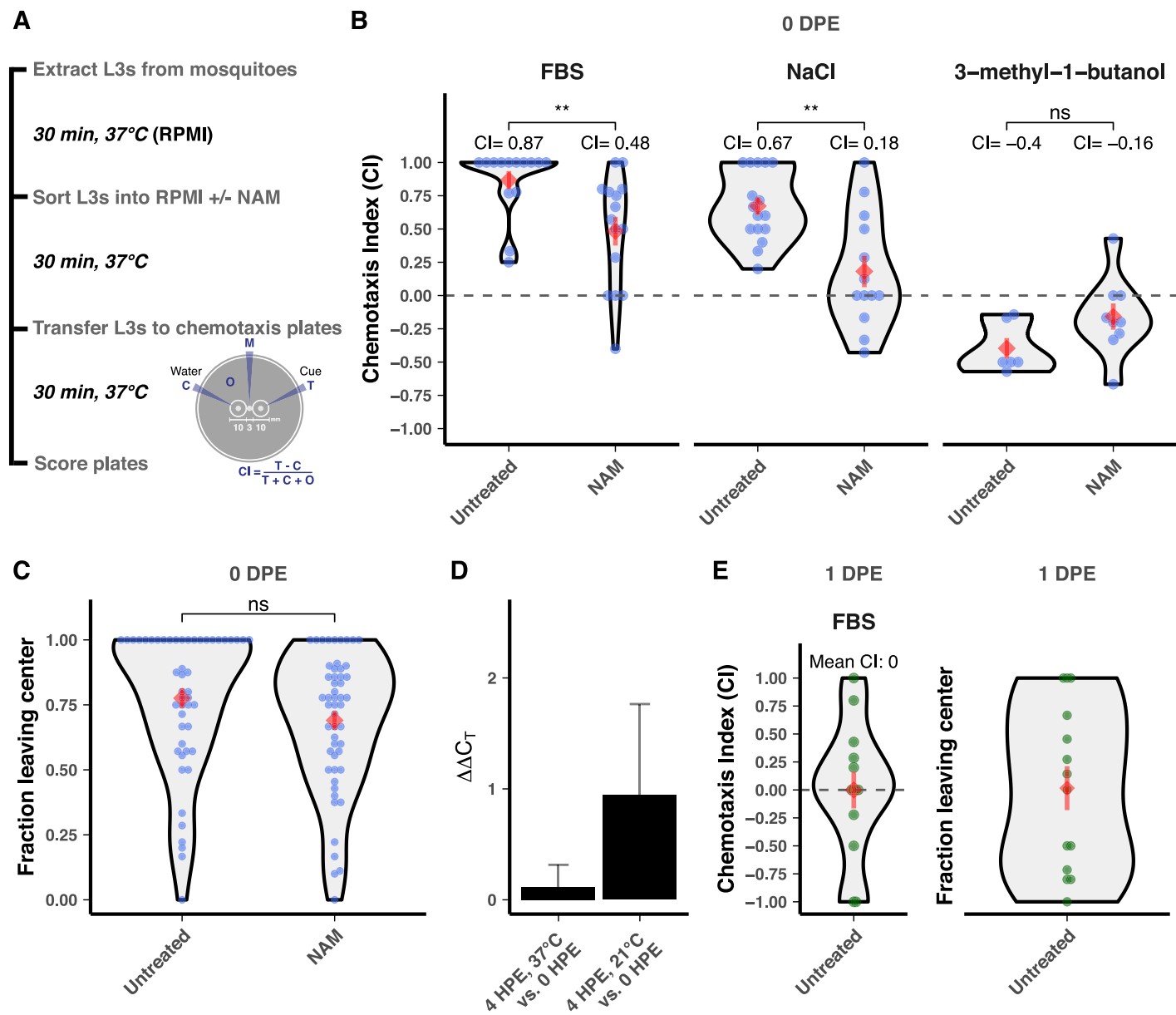

**Fig 4. Treatment with a TRPV agonist dysregulates chemotaxis of *B. pahangi* infective larvae.** (A) L3 parasites were extracted from mosquitoes and subjected to chemotaxis assays with or without 250 μM NAM treatment. Chemotaxis assays were performed by adding L3s to the middle of a 0.8% agarose plate (M), with either test cue (T) or water (C) added to the opposite sides of the plate. The plate was placed at 37°C for 30 minutes, scored after incubation, and the CI was calculated. (B) NAM dysregulates attraction of freshly extracted L3s to serum and NaCl but has no effect on aversion to 3-methyl-1-butanol. (C) NAM has no effect on translational movement of freshly extracted L3s. (D) *Bpa-osm-9* expression is unchanged by in vitro culture at physiological or room temperature 4 HPE. (E) L3s cultured for 1 DPE do not show chemotaxis toward serum and have reduced motility on the chemotaxis plate when compared with untreated freshly extracted parasites (*p* = 0.028, *t* test). Data for (A–C) represent the combined results of three independent biological replicates, except for the experiments with 3-methyl-1-butanol, which included two replicates (cohorts of mosquito infections). Data for (E) represent the results of two biological replicates. Each point represents a single chemotaxis plate with 8–10 L3s. Red diamonds and bars indicate the mean and standard error of the mean. Comparisons of means were performed using *t* tests (**\*\****p* ≤ 0.01). Raw data for (B) through (E) can be found at https://github.com/zamanianlab/BrugiaChemo-ms. CI, chemotaxis index; DPE, days postextraction; FBS, fetal bovine serum; HPE, hours postextraction; L3, third stage larvae; NAM, nicotinamide; ns, not significant; TRP, transient receptor potential.

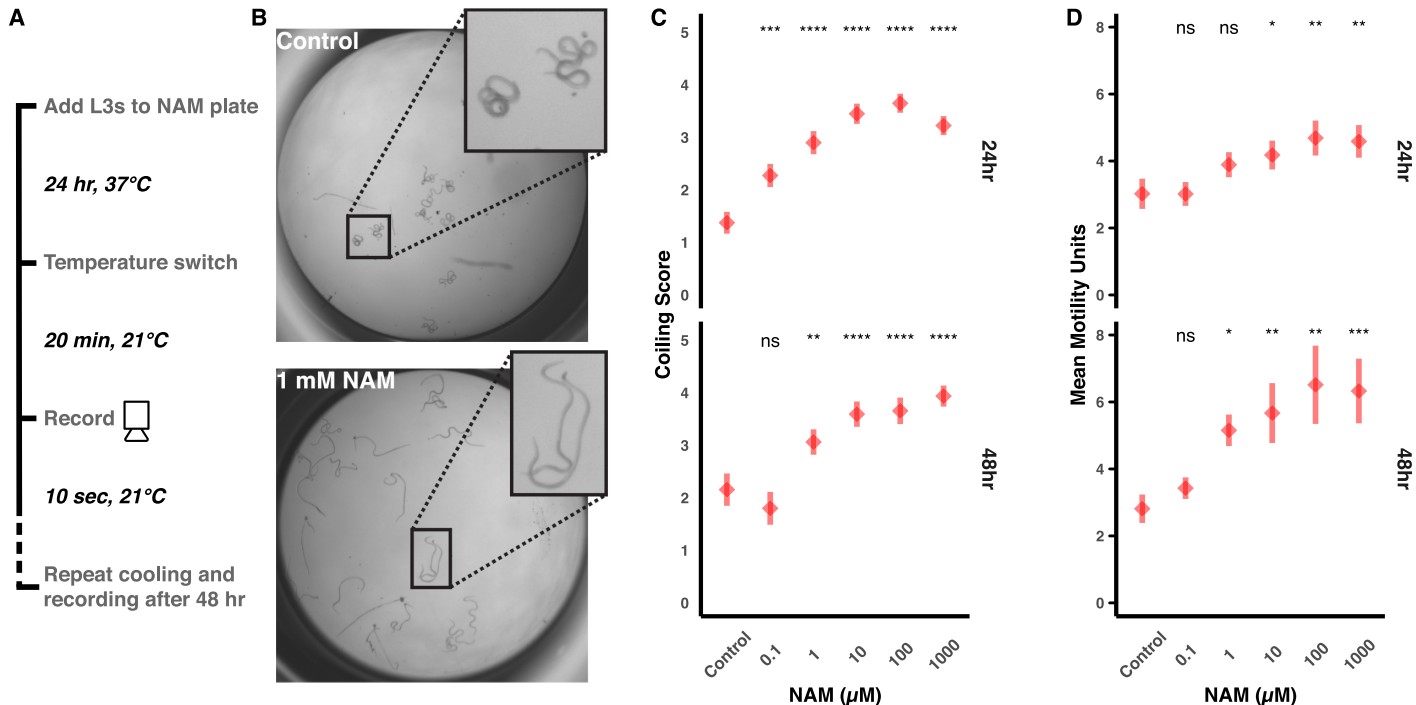

**Fig 5. Treatment with a TRPV agonist impairs the coiling response in cooled *B. pahangi* infective larvae.** (A) L3s were extracted from mosquitoes and treated with NAM, subjected to a temperature shift, and analyzed for cooling-induced coiling behaviors. (B) Representative images of untreated (control) individuals displaying the coiled phenotype and individuals exposed to 1 mM NAM that are uncoiled and thrashing. (C) Blinded coiling score given to each treatment after 24 hours and 48 hours posttreatment (higher score indicates less coiling). (D) Mean motility calculated by an optical flow algorithm. Red diamonds and bars indicate the mean and standard error of the mean from three biological replicates, each composed of >3 technical replicates scored by three different researchers. Comparisons of means were performed using one-sided *t* tests (*$p \leq 0.05$; **$p \leq 0.01$; ***$p \leq 0.001$; ****$p \leq 0.0001$). Raw data for (C) and (D) can be found at https://github.com/zamanianlab/BrugiaChemo-ms. L3, third stage larvae; NAM, nicotinamide; ns, not significant; TRP, transient receptor potential.

and a bespoke computer imaging analysis of larval coiling reveal that NAM inhibits this thermosensory response in a dose-dependent manner after 24 and 48 hours (Fig 5B–5D). These data suggest that *Brugia* OSM-9 plays a polymodal sensory role in L3 parasites, potentially mediating both chemosensory and thermosensory responses.

## Pretreatment of mf with NAM reduces L3 burden in infected mosquitoes and alters tissue distribution

Assays with extracted L3s indicated that *Bpa*-OSM-9 is important for in vitro chemoattraction to salt and serum. We hypothesized that NAM could dysregulate intramosquito chemotaxis of larval stages in vivo. To establish an assay to test this hypothesis, we first investigated whether NAM had any effect on mosquito blood-feeding dynamics. The presence of NAM in defibrinated sheep's blood altered the feeding behavior of *Ae. aegypti* when offered ad libitum on a membrane feeder. NAM at concentrations from 0.1 μM to 5 mM acted as a phagostimulant, causing a dose-dependent increase in the proportion of mosquitoes that had fed after 30 minutes. However, concentrations greater than 5 mM began to decrease the proportion of feeding individuals, and blood with 250 mM NAM was completely repulsive to mosquitoes (Fig 6A). We chose 5 mM and 25 mM as initial treatment concentrations for mf, and with replication we found that both 5 mM and 25 mM NAM caused a significant increase in the proportion of mosquitoes that fed (Fig 6B). To ensure that the increase in proportion of fed mosquitoes was not correlated to an increased blood meal size, we measured distended abdomens of

mosquitoes after feeding on control blood or blood supplemented with 5 mM or 25 mM NAM (S5 Fig). Mosquito abdomen sizes were unchanged by NAM supplementation, assuring that altered parasite burdens after feeding would not be a function of altered numbers of ingested mf (Fig 6C).

We next supplemented microfilaremic blood with 5 mM and 25 mM NAM and tested for altered infectivity and intramosquito tissue distribution of larvae. Pretreatment with NAM of *B. pahangi* mf caused a significant, dose-dependent reduction in parasite burden at 14 DPI (Fig 6D and 6E; 5 mM = 21% reduction, 25 mM = 43% reduction) and a significant decrease in the proportion of L3s recovered in the thorax of infected mosquitoes (Fig 6F). The proportion of L3s recovered in the thorax was not correlated to total L3s recovered per mosquito (S6 Fig), suggesting that changes in larval infectivity were not due to differences in blood meal size but, instead, the specific action of NAM upon the parasite. Thus, we postulate that NAM inhibits the initial migration of mf from the blood bolus but that, once across (and presumably relieved of NAM exposure in the midgut), developed L3 larval parasites are able to migrate to the head at the same proportion as untreated controls.

## *Brugia osm-9* and *tax-4* RNA interference inhibits chemoattraction of infective-stage larvae toward host-associated cues

NAM is an agonist of the *C. elegans* TRPV heteromer OSM-9/OCR-4 and *Drosophila* orthologs Nanchung/Inactive when expressed in *Xenopus* oocytes, but not of either *C. elegans* subunit alone [57]. Given conservation of NAM–receptor interactions across these phyla, we expect *Brugia* OSM-9 orthologs to also respond to NAM. However, the pharmacology and subunit interactions of *Brugia* OSM-9 may differ (e.g., filarial parasites do not have a homolog of *ocr-4* [Fig 3A]), and NAM is an endogenous metabolite in *C. elegans* that has pleiotropic effects [57,59–63]. This compelled us to use a genetic approach to more directly test whether *Brugia* OSM-9 and TAX-4 are involved in parasite chemotaxis behavior.

We carried out intramosquito ("in squito") RNA interference (RNAi) [64] of both *Bpa-osm-9* and *Bpa-tax-4* in larval stages and measured the effects on L3 in vitro chemotaxis. Infected mosquitoes were injected with double-stranded RNA (dsRNA) targeting transcripts of interest at 9 DPI, corresponding to the expected timeline of the L2-to-L3 transition in the thoracic musculature [8] (Fig 7A). We attempted to confirm knock-down of target transcripts with quantitative PCR (qPCR), but the low target abundance of *Bpa-osm-9* and *Bpa-tax-4* relative to housekeeping genes, coupled with limited recovery of RNA from a small number of parasites, prevented reliable amplification. Targeting either *Bpa-osm-9* or *Bpa-tax-4* using the in squito RNAi protocol resulted in the inhibition of *B. pahangi* L3 in vitro chemotaxis toward serum at 14 DPI (Fig 7B), whereas injection of nonspecific (*lacZ*) dsRNA had no effect on chemotaxis (control chemotaxis index [CI]: 0.776, *Bpa-osm-9(RNAi)* CI: 0.360, *Bpa-tax-4(RNAi)* CI: 0.008). dsRNA treatment did not have any effect on general parasite motility on the assay plate (Fig 7C). To our knowledge, this is the first time that either *tax-4* or *osm-9* has been shown to have a specific function in chemosensation in a parasitic nematode of animals, though *tax-4* has been shown to be involved in chemotaxis in plant-parasitic nematodes [65].

## Homologous TRP and CNG channels from *B. malayi* partially rescue sensory defects in *C. elegans*

To further explore the sensory functions of *Brugia osm-9* and *tax-4*, we tested whether these genes could rescue behavioral defects in *C. elegans* strains with loss-of-function mutations in endogenous *osm-9* or *tax-4*. Although the assembled genome of *B. malayi* is nearly chromosome scale, many of the gene models remain fragmented and unconfirmed, so we performed

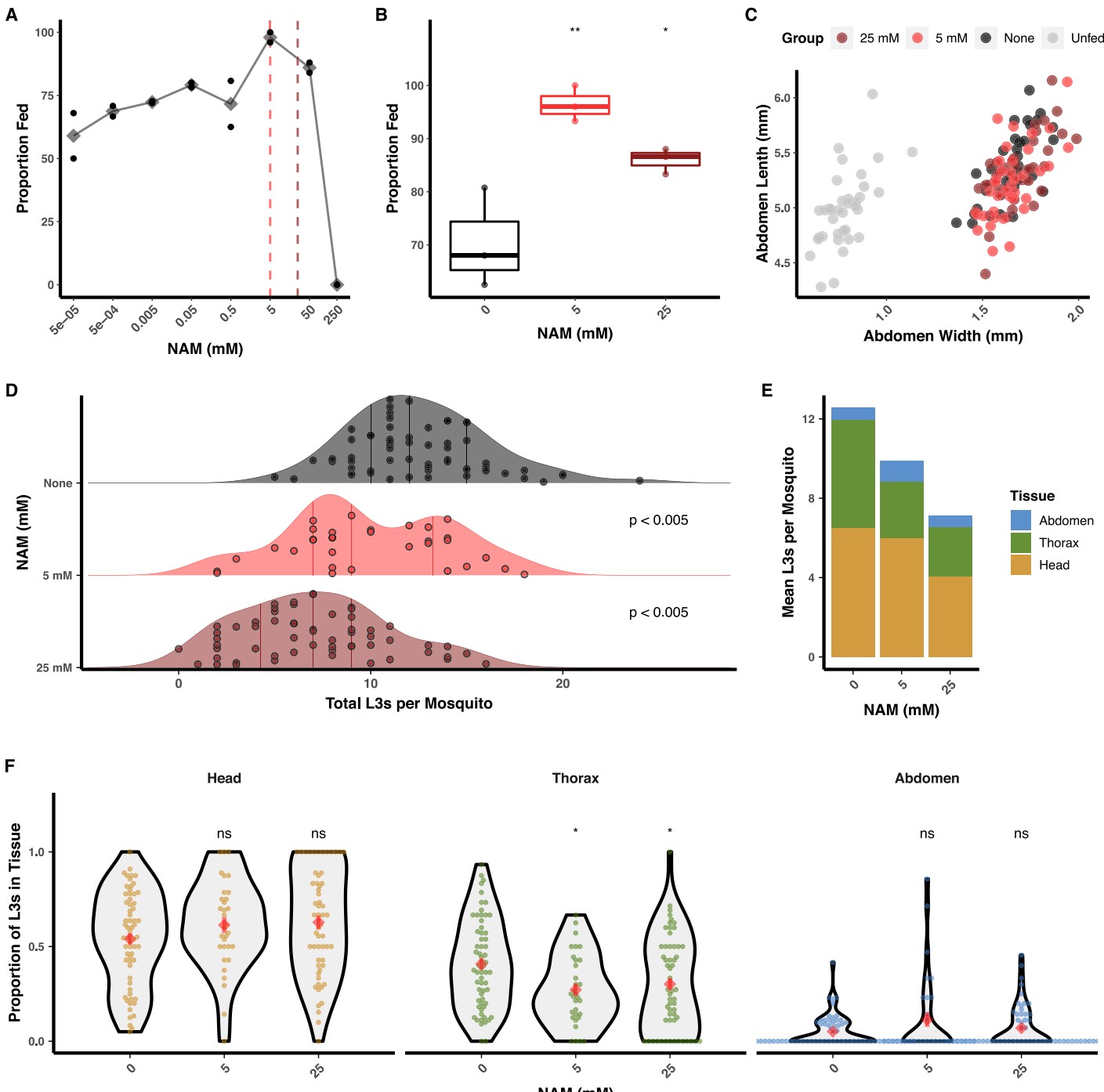

**Fig 6. Treatment with a TRPV agonist reduces the ability of mf to establish infection in mosquitoes.** (A) NAM added to blood at up to 50 mM increases the proportion of blood-fed mosquitoes when allowed to feed to repletion but reduces mosquito blood feeding at concentrations greater than 50 mM. Black points represent technical replicates, and gray diamonds represent the mean. (B) Replication of blood-feeding experiments with 5 mM and 25 mM showed a significant increase in the proportion of blood-fed mosquitoes when blood was supplemented with 5 or 25 mM NAM. These concentrations were used for subsequent parasite treatment. Points represent the values from three independent biological replicates (cohorts of mosquitoes). (C) Blood supplemented with 5 or 25 mM NAM does not alter the size of distended mosquito abdomens after blood feeding, indicating an unaltered size of blood meal. Points represent the measured abdomens of individual mosquitoes from a single blood-feeding experiment. (D-E) Pretreatment of *B. pahangi* mf with NAM prior to mosquito infection causes a dose-dependent reduction in the number of L3s recovered per mosquito. (F) Reduction in L3 recovery was due to a decrease in larval parasites in the mosquito thorax (i.e., the flight muscles, the migratory destination for mf and site of development for L1, L2, and early-L3 parasites). Data from (D-F) represent the combined results of three independent biological replicates (cohorts of mosquito infections); each point represents the parasites recovered from an individual mosquito. Red diamonds and bars indicate the mean and standard error of the

mean. Tests of significance for (B), (D), and (F) were performed with Tukey's post hoc tests and adjusted for multiple comparisons (*$p \leq 0.05$; **$p \leq 0.01$). Raw data can be found at https://github.com/zamanianlab/BrugiaChemo-ms. L1, first stage larvae; L2, second stage larvae; L3, third stage larvae; mf, microfilaria; NAM, nicotinamide; ns, not significant; TRP, transient receptor potential.

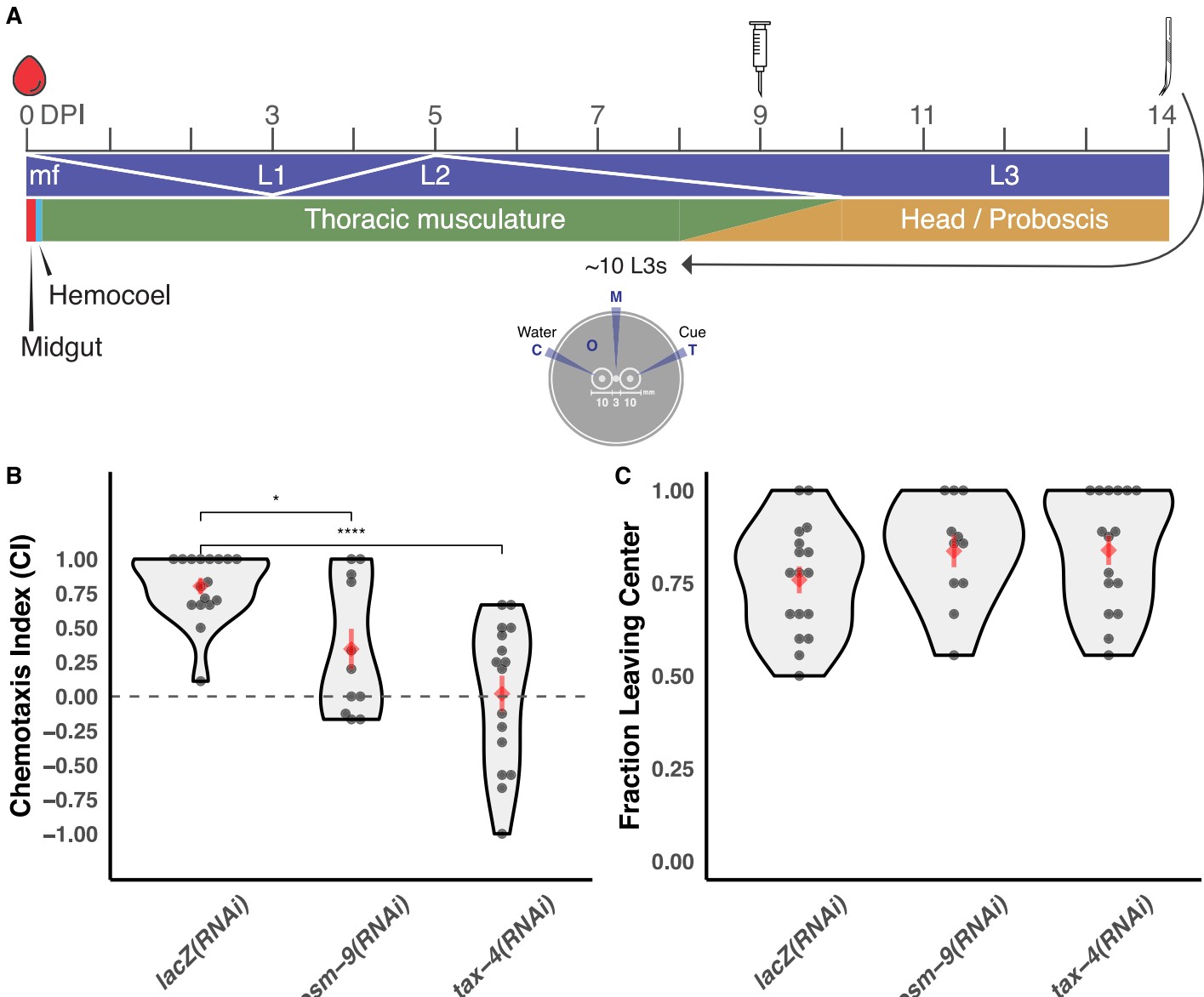

**Fig 7. dsRNA treatment of chemosensory pathway receptors causes defective chemotaxis of *B. pahangi* infective larvae.** (A) Injection of 250 ng dsRNA into *B. pahangi*-infected *Ae. aegypti* LVP was performed 9 DPI, and L3 parasites were recovered via dissection at 14 DPI. Recovered parasites were immediately used in chemotaxis experiments. Chemotaxis assays were performed by adding L3s to the middle of a 0.8% agarose plate (M), with either test cue (T) or water (C) added to the opposite sides of the plate. The plate was placed at 37°C for 30 minutes, scored after incubation, and the CI was calculated. Intramosquito developmental dynamics were adapted from [8]. (B) dsRNA treatment of *Bpa-osm-9* or *Bpa-tax-4* resulted in a reduced ability of L3s to migrate to serum. Control parasites were recovered from mosquitoes injected with *lacZ* dsRNA. (C) dsRNA exposure does not inhibit general translational motility on the chemotaxis plate. Data represent the combined results of three independent biological replicates (cohorts of mosquito infections); each point represents the CI of an individual plate. Red diamonds and bars indicate the mean and standard error of the mean. Comparisons of means were performed using *t* tests (*$p \leq 0.05$; ****$p \leq 0.0001$). Raw data can be found at https://github.com/zamanianlab/BrugiaChemo-ms. CI, chemotaxis index; DPI, days postinfection; dsRNA, double-stranded RNA; L1, first stage larvae; L2, second stage larvae; L3, third stage larvae; LVP, Liverpool strain; mf, microfilaria.

low-coverage isoform sequencing with long-read RNA-seq on *B. malayi* adult males and females. This led to the successful capture of *Bma-osm-9* full-length transcripts, but it failed to capture *Bma-tax-4*. Using these data and the predicted gene model of *Bma-tax-4*, we cloned these genes for functional expression in *C. elegans*. We also cloned the *Bma-ocr-1/2*–like gene (*Bm5691*, or *Bma-ocr-1/2a*) that has the highest predicted amino acid identity to OCR-2, which functions with OSM-9 in *C. elegans* to enable a range of sensory behaviors.

The *Bma-osm-9* clone and the two full-length isoforms captured by long-read sequencing included a 41-bp insertion that corresponded to a missing splice acceptor site at intron 17 that was not reflected in the original gene prediction (S7 Fig). This insertion caused a frameshift in the predicted amino acid sequence that made the resulting sequence more similar to the *Cel-osm-9* sequence than the original prediction (S8 Fig). The consensus *Bma-tax-4* transcript we cloned had two differences from the predicted model: a synonymous 694T>C that was found in four out of seven sequenced clones and a 21-bp deletion that was found in all clones and corresponds to a mispredicted splice donor site at intron 2 (S9 Fig). The consensus *Bma-ocr-1/2a* sequence was a perfect match to the predicted gene model. We used these clones and an array of sensory assays to test for rescue of *C. elegans* sensory defects by the *B. malayi* homologs.

These genes were initially expressed in corresponding loss-of-function *C. elegans* backgrounds (*osm-9(ky10)* and *tax-4(p678)*) using *Cel-osm-9* and *Cel-tax-4* promoter regions [30,32] and the *unc-54* 3′ UTR, which is commonly used for expression in somatic cells [66,67]. Though transcripts were captured via qPCR of RNA from whole worm homogenates (S10 Fig), all *osm-9* transgenes (including the endogenous open reading frame) were unable to rescue chemotaxis (S11 Fig). Coexpression of *Bma-ocr-1/2a* with *Bma-osm-9* also did not enable rescue of chemotaxis (S11 Fig). We subsequently replaced the *unc-54* 3′ UTRs with 3.2 kb of the *Cel-osm-9* downstream region and found that this modification allowed for partial rescue of the avoidance defects but not of chemotaxis defects (Fig 8A and 8B, S12 Fig). Self-

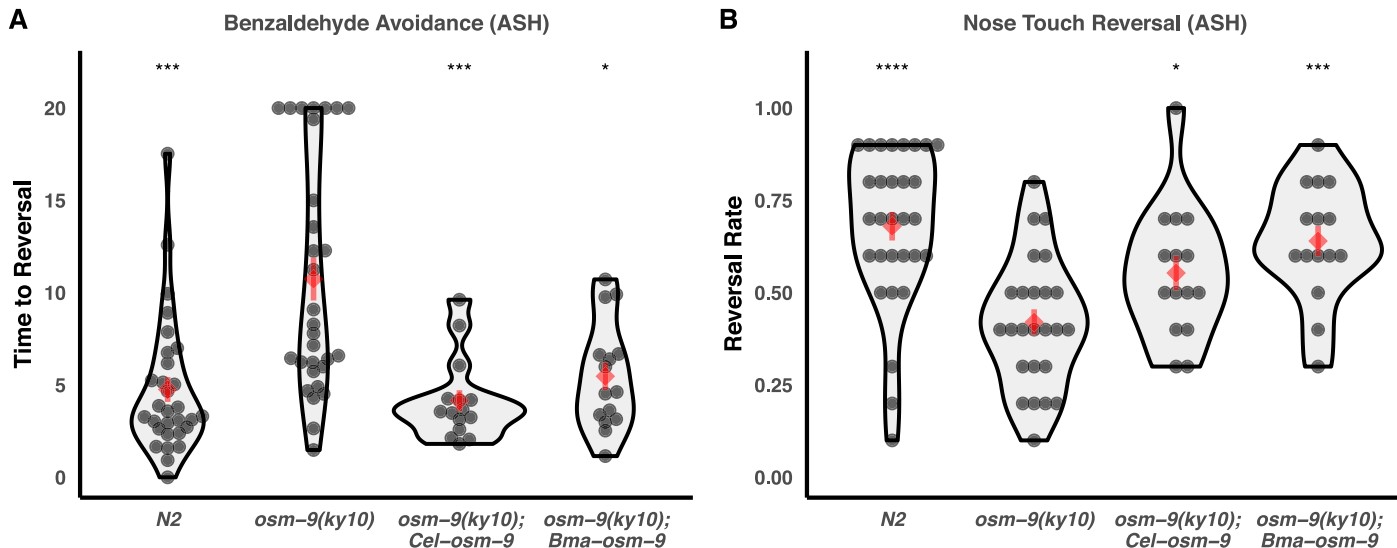

**Fig 8. Heterologous expression of *B. malayi osm-9* partially rescues loss-of-function sensory defects in *C. elegans*.** *Bma-osm-9* was cloned and expressed under the control of the endogenous *Cel-osm-9* promoter and 3′ UTR [30,32]. The *Cel-osm-9* open reading frame was used as a positive control. Avoidance defects (OSM-9 functioning in ASH) to (A) concentrated benzaldehyde or (B) mechanical nose touch were partially rescued by the positive control and by *Bma-osm-9*. Data represent the combined results of at least three independent biological replicates, each consisting of five technical replicates. Each point represents the recorded value of an individual worm. Comparisons to the loss-of-function strain were performed using *t* tests (*$p \leq 0.05$, ***$p \leq 0.001$, ****$p \leq 0.0001$). Raw data can be found at https://github.com/zamanianlab/BrugiaChemo-ms.

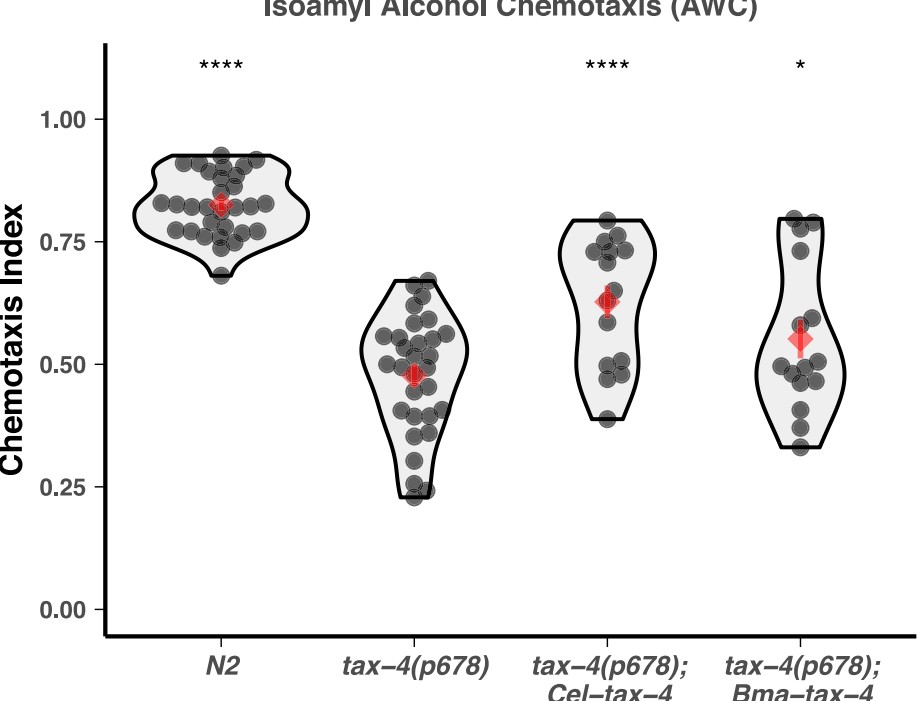

**Fig 9. Heterologous expression of *B. malayi tax-4* partially rescues loss-of-function chemotaxis defects in *C. elegans*.** *Bma-tax-4* was cloned and expressed under the control of the endogenous *Cel-tax-4* promoter. The *Cel-tax-4* open reading frame was used as a positive control. Both constructs partially rescued the chemotaxis defect to isoamyl alcohol (TAX-4 functioning in AWC). Comparisons to the loss-of-function strain were performed using *t* tests (*$p \leq 0.05$, ****$p \leq 0.0001$). Raw data can be found at https://github.com/zamanianlab/BrugiaChemo-ms.

rescue with *C. elegans osm-9* with the endogenous 3′ UTR also did not rescue the chemotaxis defect, suggesting that there are additional regulatory elements that enable expression in AWA. *Bma-tax-4* showed partial rescue of chemotaxis to isoamyl alcohol, which is controlled by the AWC neuron, even without *Cel-tax-4* 3′ *cis*-regulatory elements (Fig 9). These results show that *Brugia* OSM-9 and TAX-4 are able to partially rescue *C. elegans* sensory defects by functioning as channel subunits in a free-living nematode cell context, suggesting some functional conservation across diverged nematode species.

## Discussion

Filarial worms continue to pose a significant threat to human and animal health. The ability of filarial worms to move through and between hosts relies on their ability to sense their environment, evidenced by the diversity of genus-specific niches occupied by different filariae when they migrate within shared arthropod or vertebrate hosts (e.g., *Brugia* larvae migrate to the thoracic musculature of *Ae. aegypti*, whereas *Dirofilaria* migrate to the Malphigian tubules). Despite the importance of sensory behaviors in the evolution and persistence of parasitism, little is known of the receptors and pathways that control such behaviors in parasitic nematodes. Identifying mediators of sensory-associated behaviors can aid our understanding of disease transmission and pathogenesis and may also provide new targets for therapeutic intervention.

We have shown that filarial worms have a greatly reduced and divergent set of chemoreceptors as compared with *C. elegans* and the rest of the Nematoda but that they retain much of the core downstream chemosensory signaling pathway. Filarial parasites exhibit stage-, tissue-,

and sex-specific chemoreceptor patterns that likely correspond to the different vector and host environments encountered throughout the life cycle. We expect that these patterns correspond to landmark migration events, including the migration of larvae within the mosquito host, the transmission of infective larvae (L3s) to the definitive host, the early migration of larvae in the definitive host, and the potential mate-seeking behaviors of dioecious adults.

Historically, it has been difficult to identify endogenous ligands for nematode chemoreceptors. In *C. elegans*, only a small fraction of chemoreceptors have been linked to activating molecules, some of which are unlikely to be the natural ligand [50,68–77]. This is partly a function of the large number of chemoreceptors (>1,400) and the large space of potentially relevant terrestrial cues in free-living clade V nematodes. The smaller complement of chemoreceptors in clade IIIc filarial parasites combined with the stark temporal receptor expression patterns overlayed with possible host-derived molecules present at intrahost sites of parasite migration may facilitate a comparatively easier path to chemoreceptor deorphanization. Efforts are underway to develop new heterologous expression platforms for deorphanization that may be more amenable to the expression of nematode GPCRs, which have often been recalcitrant to expression in single-cell mammalian or yeast systems.

As in *C. elegans*, TRP and CNG channels in filarial worms likely function downstream of chemoreceptors expressed on the cilia of sensory amphids. We show that a TRPV chemical agonist inhibits in vitro chemotaxis of infective *Brugia* larvae toward host-associated cues and compromises the ability of mf to establish mosquito infections. RNAi experiments implicate both *Brugia* OSM-9 and TAX-4 as necessary for larval chemotaxis to serum. In *C. elegans*, OSM-9 and TAX-4 function as sensory transducers in distinct sensory cells that can respond to distinct cues. It is interesting that both OSM-9 and TAX-4 mediate responses to serum in *Brugia*. FBS is a complex heterogeneous mixture of macromolecules, amino acids, and ions, and it is conceivable that different components of this mixture activate distinct chemoreceptors, chemosensory neurons, and downstream pathways in *Brugia*. Little is known about the neuronal architecture of filarial worms, and it is also possible that OSM-9 and TAX-4 are coexpressed and have homologous function in the same cells. A map of the filarial worm connectome and the ability to produce transcriptional reporters would help resolve this question and will be hastened by technologies to more easily dissect filarial worm neuroanatomy and neurogenetics [78]. Furthermore, fractionation of serum into pure chemicals for chemotaxis experiments could illuminate whether multiple cells are involved in *Brugia* chemotactic responses to serum or other crude preparations.

CNG and TRP channels are polymodal in *C. elegans* and function in sensory neurons responsible for aerosensation, mechanosensation, chemosensation, noxious avoidance, and thermosensation, among others [30–32,79–81]. Our data suggest a similar polymodal deployment of these channels in mosquito-borne filarial worms, though the pattern of neuronal expression may differ. Although OSM-9 in *C. elegans* is involved in noxious heat avoidance in the nociceptive ASH neuron [80], it is TAX-4 and TAX-2 that function in the sensation of precise thermal gradients via the AFD neuron. Whether filarial worms have cooperative thermal sensory programs is unknown, but given the range of temperatures experienced by these parasites (from ambient temperatures while in mosquitoes to physiological temperatures in mammalian hosts) and the unlikelihood of experiencing or being able to avoid noxious heat or cold, the sensory program in filarial worms is likely more simple than their free-living counterparts. Our data suggest that OSM-9 is involved in this program, but CNG channels like TAX-4/TAX-2 cannot be ruled out.

Questions remain as to the stoichiometry and subunit interactions of TRP and CNG channels that function in *Brugia* chemosensation. Both *Bma-osm-9* and *Bma-tax-4* were able to rescue sensory defects in *C. elegans* without coexpression of putative subunits (e.g., *ocr-2* and *tax-*

*2*), so it is possible that the parasite subunits were able to form heteromeric channels with their free-living counterparts or that the parasite subunits were able to form homomeric channels. Regardless, the clade IIIc loss of CNG channels involved in olfactory plasticity (*cng-1*, *cng-3*) and TRP channels that are expressed in the mechanosensory labial QLQ neurons of *C. elegans* (*ocr-4*, *trpa-1*) demonstrate imperfect conservation of all sensory modalities and pathways between *C. elegans* and filarial worms, and it is possible that filarial worm TRP and CNG channels have evolved subunit interactions or primary functions that are not conserved in *C. elegans* [33,54,82,83].

Deeper knowledge of chemotaxis has been achieved in clade IV and V nematodes, which are more amenable than filarial worms to in vitro culture and manipulation [16,84]. Continued development of genetic tools [64,78,85] and sensory assays would help to further elucidate the molecular basis for sensory behaviors in clade III LF parasites. The in vitro chemotactic capacity of filarial worm L3s was transitory under our assay conditions. Adults, which are healthy in culture much longer than larval parasites, and mf, which can be generated in greater numbers, could offer additional platforms for sensory pathway dissection. Exploration of other sensory modalities, such as thermosensation and mechanosensation, is essential to develop a more thorough model of how filarial worms integrate sensory data in order to successfully invade, infect, and migrate within the host.

## Materials and methods

### Parasite and mosquito maintenance

mf, L3, and adult-stage FR3 strains of *B. malayi* and *B. pahangi* from the NIH/NIAID Filariasis Research Reagent Resource Center (FR3) [86] were maintained in RPMI 1640 culture media (Sigma-Aldrich, St. Louis, MO) with penicillin/streptomycin (0.1 mg/mL; Gibco, Gaithersburg, MD) and FBS (10% v/v; Gibco) unless otherwise stated. For local production of L3 *B. pahangi*, mf were incubated in defibrinated sheep's blood (Hemostat, Dixon, CA) at a density of 120–160 mf per 20 µL at 37˚C provided via a membrane feeder [87]. mf were exposed to groups of 250 adult female (1–3 days postemergence) *Ae. aegypti* LVP mosquitoes, which had been starved 1 day prior to feeding. Infected mosquitoes were maintained in double-caged cartons in a Percival Scientific incubator (I-36NL, Perry, IA) at 26˚C with 85% humidity and a 12-hour light/dark cycle and provided 10% sucrose throughout.

At 14 DPI, L3 parasites were extracted into warm *Aedes* saline [88] or RPMI 1640 via microdissection of cold-anesthetized mosquitoes or bulk isolation as previously described [9]. Prevalence and locality of L3s were determined by separating head, thorax, and abdominal tissues during dissection with all L3s counted per mosquito. *B. malayi* adults used for RNA-seq were received from the FR3 and were immediately washed and placed in new complete media. Adult parasites were allowed to equilibrate at 37˚C for 24 hours before any further experimentation.

### *C. elegans* strains

*C. elegans* strains were maintained at 20˚C on NGM plates seeded with *Escherichia coli* strain OP50 and routinely picked to fresh plates at the L4 stage. Transgenic strains were created as described [89] by microinjecting 50 ng/µL of parasite transgene, combined with either *unc-122p::GFP* or *myo-2p::GFP* as coinjection markers and an empty vector to achieve a final concentration of 100 ng/uL. Three independently derived lines were created for each transgenic strain and maintained separately. Genotypes used include *osm-9(ky10)* IV, *tax-4(p678)* III, ZAM13: *osm-9(ky10)* IV, *mazEx13*[*osm-9p::Bma-osm-9::unc-54* 3′UTR*; unc-122p::GFP*], ZAM14: *tax-4(p678)* III, *mazEx14*[*tax-4p::Bma-tax-4::unc-54* 3′UTR*; myo-2p::GFP*], ZAM17:

*osm-9(ky10)* IV, *mazEx13[osm-9p::Bma-osm-9::unc-54 3′UTR; unc-122p::GFP]*, *mazEx17[osm-9p::Bma-ocr-1/2a::unc-54 3′UTR; myo-2p::GFP]*, ZAM18: *osm-9(ky10)* IV, *mazEx18[osm-9p::Cel-osm-9::unc-54 3′UTR; unc-122p::GFP]*, ZAM21: *tax-4(p678)* III, *mazEx19[tax-4p::Cel-tax-4::unc-54 3′UTR; myo-2p::GFP]*, ZAM22: *osm-9(ky10)* IV, *mazEx20[osm-9p::Bma-osm-9::osm-9 3′UTR; myo-2p::GFP]*, ZAM24: *osm-9(ky10)* IV, and *mazEx21[osm-9p::Cel-osm-9::osm-9 3′UTR; myo-2p::GFP]*.

mRNA expression of some *B. malayi* genes in transgenic *C. elegans* was confirmed with qPCR (S10 Fig; primer sequences are included in S4 File). A 20-μL qPCR reaction was optimized using 2× PowerUp SYBR Green MasterMix (Thermo Fisher Scientific) and 10 ng of RNA isolated from mixed populations from two to three chunked plates as described [90] converted to cDNA with SuperScript III using an equal amount of random hexamer and oligo (dT) primers. Reactions were run in duplicate on a StepOnePlus real-time PCR system (Applied Biosystems, Waltham, Massachusetts). $C_T$ values were calculated with the system's automatic threshold, and relative expression was calculated with the $\Delta\Delta C_T$ method [91]. Parasite transcripts in negative controls (knock-out *C. elegans* strains) were undetected in all cases, and $C_T$ was set to 40 to calculate a $\Delta\Delta C_T$ for data visualization.

### Comparative genomics

**Chemosensory GPCRs.** The chemoreceptor mining and annotation strategy is charted in S1 Fig. Briefly, all filarial worm predicted proteomes in WormBase ParaSite version 9 [39] and a selected list of high-quality genomes that included representatives from the four major nematode clades (39 total species, S1 Table) were searched (hmmsearch [92]) against a database of profile hidden Markov models (HMMs) curated by Pfam, consisting of primary metazoan GPCR families and the nematode chemoreceptor families [93]. Predicted proteins were filtered such that each that had a best-hit to a nematode chemoreceptor HMM was retained. Surviving predicted proteins were then used in a reciprocal search (hmmsearch) against the entire Pfam HMM database. Predicted proteins that had a best-hit to a nematode chemoreceptor HMM were retained. Surviving predicted proteins were then searched (blastp [94]) against the *C. elegans* predicted proteome (N2, WBPS9), and predicted proteins that had a best-hit to a *C. elegans* chemoreceptor (S4 Data) were retained (*C. elegans* chemoreceptors were obtained by downloading all named chemoreceptors from WormBase and their paralogues via the WormBase ParaSite API).

**TRP and CNG receptors.** Predicted protein sequences of annotated TRP and CNG channels from *C. elegans* were downloaded from WormBase [95] and used as seeds in blastp searches against all predicted proteomes included in S1 Table. Hits with an E-value < 0.01 were reciprocally searched against the *C. elegans* predicted proteome, and any hit that was not most similar to a *C. elegans* TRP or CNG channel was removed. Because there were clade and species-specific gene losses in the CNG family, *C. elegans* seeds were also used in a tblastn search against parasite genomes to account for missing gene models and possible errors in gene predictions.

### Phylogenetics

**Chemosensory GPCRs.** Predicted protein sequences belonging to *C. elegans* chemoreceptor families [40] were aligned by family with MAFFT [96]. The resulting family profile HMMs were sequentially aligned with MUSCLE [97] to create a master *C. elegans* chemoreceptor alignment. Predicted chemoreceptors from 19 selected species underwent transmembrane (TM) domain prediction with HMMTOP [98], and only those that contained exactly 7 predicted TMs were aligned to the master alignment. This final multiple sequence alignment was

trimmed with trimAl [99] such that columns with greater than 30% gaps were removed, and sequences that did not have at least 70% of residues that aligned to columns supported by 70% of the sequences were removed.

The trimmed, filtered alignment was subjected to maximum-likelihood phylogenetic inference with IQ-TREE [100] and ModelFinder [101] with ultrafast bootstrapping [102], using the VT substitution matrix [103] with empirical base frequencies and a free-rate substitution model [104,105] with 10 categories. Bootstrap values from 1,000 replicates were drawn as nodal support onto the maximum-likelihood tree.

**TRP and CNG receptors.** Putative TRP sequences underwent TM prediction, and any sequence with $\geq 1$ predicted TM was retained. TRP and CNG sequences were separately aligned and trimmed such that columns with greater than 25% gaps were removed, and CNG sequences that did not have at least 70% of residues that aligned columns supported by 70% of the sequences were removed. For both datasets, fragments with large gaps or putative isoforms were manually removed. Alignments were subjected to Bayesian phylogenetic inference with MrBayes [106]. The MCMC chain was run for 10,000,000 generations, with a sample taken every 1,000 generations. Eight separate chains were run, with two hot chains and the temperature set to 0.05. Consensus trees were drawn using the 50% majority rule, with all compatible groups added, and posterior probabilities were drawn as nodal support. All trees were annotated with ggtree [107].

## *B. malayi* transcriptomic analyses

**Anterior and posterior B. malayi transcripts.** One millimeter of the anterior and posterior ends of 19 adult male and 18 adult female *B. malayi* were cut from live parasites and immediately transferred to Trizol (Ambion, Waltham, MA). Tissue in Trizol was homogenized with a plastic pestle, and RNA was extracted with Direct-zol RNA miniprep kit (Zymo, Irvine, CA) according to the manufacturer's instructions and was eluted in RNase-free water. RNA was DNase treated on the column, and the quality of purified RNA was assessed with Qubit (Thermo Fisher Scientific, Waltham, MA) and Bioanalyzer Pico chip (Agilent, Santa Clara, CA). RNA was rRNA depleted with Ribo-Zero ScriptSeq Complete Gold (Blood) (Illumina, San Diego, CA), and sequencing libraries were constructed using the TruSeq Stranded Total RNA kit (Illumina). All samples were sequenced at the University of Wisconsin-Madison Biotechnology Center with an Illumina HiSeq 2500 with a single-end 100-bp read setting. Reads were adapter and quality trimmed using Trimmomatic [108]. HISAT2 [109] and String-Tie [110] were used to align reads to the *B. malayi* reference genome (WormBase ParaSite [39], release 12 version 4) and to produce TPM counts for annotated genes. The RNA-seq pipeline was implemented using Nextflow [111] and is publicly available (https://github.com/zamanianlab/Bmalayi_HTRNAseq-nf). Custom R scripts were used for profiling, hierarchical clustering, and visualization of putative chemosensory gene expression across anterior and posterior samples.

**Stage-specific expression of B. malayi chemosensory genes.** Public stage-specific RNA-seq data [53] were acquired from NCBI SRA. Reads were aligned to version 4 of the *B. malayi* genome, which was downloaded from version 12 of WormBase ParaSite [39]. Reads were aligned with HISAT2 [109] and StringTie [110]. Custom R scripts were used for profiling, hierarchical clustering, and visualization of putative chemosensory gene expression across life stages. Heatmaps of life stage expression were drawn according to chromosomal location. The RNA-seq pipeline was implemented using Nextflow [111] and is publicly available (https://github.com/zamanianlab/BmalayiRNAseq-nf). Locus information was extracted from GTF files from WormBase ParaSite[39][112].

## *Brugia* chemotaxis assays

All chemotaxis assays were performed immediately after extraction of L3s from in-house infections and following previously published protocols [25–27]. For NAM (DOT Scientific, Burton, MI) treatment experiments, extracted parasites were first sorted from warm RPMI 1640 into room temperature RPMI 1640, and half of the parasites were placed in media supplemented with a final concentration of 250 μM NAM and incubated for 30 minutes. Heat-inactivated FBS (Gibco), 1 M NaCl (Thermo Fisher Scientific), and 1:1 3-methyl-1-butanol (Thermo Fisher Scientific) in mineral oil were used as cues. A curved platinum worm pick was used to remove L3s from warm media and place them on 0.8% agarose plates; plates were transferred to a 37°C incubator with 5% atmospheric $CO_2$; and parasites were allowed to migrate for 30 minutes, after which the plates were removed and scored. Eight to ten parasites were added per plate. The CI of each plate was calculated as follows: CI = (T − C) / (T + C + O), where T is the number of parasites that migrated to the test cue, C is the number that migrated to the control cue, and O is the number that migrated outside of the designated cue areas. To account for parasite injury in transfer, only plates that had C + T + O $\geq$ 5 were used for statistical analysis and plotting. Experiments using FBS and NaCl as cues included three biological replicates, and experiments with 3-methyl-1-butanol included two biological replicates. Biological replicates are defined as groups of larvae originating from separate cohorts of mosquito infections (approximately 250 infected mosquitoes) and separate cohorts of mf extractions. L3 parasites were subsequently extracted and assayed on different days. Technical variation was accounted for on each assay day by performing at least three assays (i.e., chemotaxis plates) per biological replicate.

## Larval coiling assay

NAM treatment of *B. pahangi* L3s was performed with parasites from in-house infections or received from the FR3. After extraction or receipt, parasites were washed with fresh RPMI 1640 and suspended in complete media (RPMI 1640 + 10% FBS + penicillin/streptomycin) at a density of one parasite per 2 μL. A 96-well plate was arranged with 50 μL of complete media with 2× NAM in each well, and parasites were pipetted into each individual well to create a density of 10–25 parasites per well in a final volume of 100 μL. Plates were immediately transferred to a 37°C incubator with 5% atmospheric $CO_2$ and were left untouched until videos were recorded at 24 and 48 hours posttreatment. Care was taken not to disturb parasites while transferring the plates from the incubator to the recording stage. Parasites were allowed to cool at room temperature for 20 minutes on the recording stage, after which each well was recorded for 10 seconds at 16 FPS. Recording was performed at 2.5× on a Zeiss Stemi 508 with a K LAB stand and oblique light with a monochrome CMOS camera (DMK 33UX178, The Imaging Source). Parasites were returned to the incubator immediately after recording.

Coiling experiments included three biological replicates, defined as groups of larvae originating from separate cohorts of mosquito infections (approximately 250 infected mosquitoes) and separate cohorts of mf extractions. L3 parasites were extracted and assayed on different days. An entire dose–response curve was performed for each replicate (100 nM to 1 mM), and at least four technical replicates (wells) of 10–25 parasites were included in each biological replicate.

Videos were manually scored and analyzed with a bespoke optical flow algorithm implemented in Python that calculates mean motility units (mmu) and has similarities to previous implementations [113,114]. Relevant differences include the utilization of a dense flow algorithm that analyzes every pixel in the image instead of focusing on a sparse set of features, post hoc analysis rather than real-time tracking to allow for greater quality control, and image

segmentation to calculate worm area to enable interwell normalization. Source code and a recommended Conda environment can be found at https://github.com/zamanianlab/BrugiaMotilityAnalysis.

For manual scoring of larval coiling, videos were assigned randomized file names and distributed to three researchers. Researchers blindly rated each well on a scale of 0–5, where 0 is the most coiling and 5 is the least coiling (the template with scoring instructions is provided in S3 File). Scores were collated, and data were blindly analyzed and plotted with the tidyverse package [115] and custom R scripts.

### *Ae. aegypti* feeding and engorgement assays

At 2–3 days postemergence, pools of 25–50 adult female *Ae. aegypti* LVP strain mosquitoes were starved for 24 hours and then provided with a blood meal for 30 minutes via a glass membrane feeder [87]. Immediately after feeding, groups were cold anesthetized and visually inspected for distended abdomens to measure the proportion of feeding mosquitoes. Blood meal size was measured by calculating the ratio of the mosquito length (abdomen tip to thorax) to width (dorsoventral width at the fifth abdominal segment) using Fiji [116](S5 Fig).

### Larval temperature-shift assay and qPCR

*B. pahangi* L3s were extracted in bulk and separated into three treatment groups: immediate storage in Trizol LS (Ambion); 1 mL RPMI 1640 + 10% FBS + penicillin/streptomycin at room temperature; or 1 mL RPMI 1640 + 10% FBS + penicillin/streptomycin in a 37°C heat block. Parasites were incubated in media for 4 hours. After incubation, medium was removed, and parasites were washed once in fresh RPMI 1640 and stored in Trizol LS at −80°C until processing. To extract RNA, samples were thawed on ice, and the volume was adjusted to a final ratio of 3:1 Trizol LS:RNase-free water. Samples were lysed with a TissueLyser LT (Qiagen, Venlo, The Netherlands). One 5-mm stainless steel bead was added to each tube, which then underwent two cycles of 3 minutes of shaking at 30 Hz. Tubes were cooled on ice for 2 minutes in between cycles. RNA was extracted with the Direct-zol RNA miniprep kit (Zymo) according to the manufacturer's instructions, including an on-column DNase treatment, and RNA was eluted in 15 μL of RNase-free water. RNA samples were quantified with a NanoDrop 1000 and immediately used for first-strand cDNA synthesis with SuperScript III (Thermo Fisher Scientific) using random hexamers and normalizing RNA input. cDNA was stored at −20°C until further use.

For qPCR, GAPDH control primers [117] and *osm-9* primers (designed with Primer3 [118], F: CCCGCTGATCCAAACATTG, R: TGCACTACACGTCATATCACTG) were optimized with *B. pahangi* L3 RNA from the FR3 with cDNA synthesized using the same SuperScript III master mix as the experimental RNA samples. A 20-μL reaction was used with 2× PowerUp SYBR Green MasterMix, 800 nM primers, and 5.2 ng RNA. Reactions were run in duplicate on a StepOnePlus real-time PCR system. $C_T$ values were calculated with the system's automatic threshold, and relative expression was calculated with the $\Delta\Delta C_T$ method [91].

### In squito exposure of infective larva to *Bpa-osm-9* and *Bpa-tax-4* dsRNAs

Primers were designed to amplify 200- to 600-bp regions from cloned *Bma-tax-4* and *Bma-osm-9*, which had >95% identity with their *B. pahangi* orthologs, and T7 recognition sequences were appended to the 5′ end of each primer. Cloned genes (below) were used as template DNA for PCRs with Phusion polymerase (New England Biolabs, Ipswich, MA). Complete dsRNA synthesis protocols, including primer sequences and thermocycler programs, can be found in S4 File. PCR product was cleaned (Qiagen MinElute PCR Purification

Kit) and resuspended in water at a desired concentration of 1–2 µg/µL as measured by a Qubit 3.0 dsDNA assay (Thermo Fisher Scientific). This product was subsequently used as the template for a dsRNA synthesis reaction (MegaScript RNAi, Thermo Fisher Scientific). dsRNA was RNase and DNase treated, purified with phenol/chloroform, precipitated in cold isopropanol, and resuspended in nuclease-free water at a concentration of 1–4 µg/µL. The concentrations and purity of 1:20 dilutions of dsRNA were measured with a NanoDrop 1000 (Thermo Fisher Scientific).

*Ae. aegypti* LVP strain mosquitoes were infected in batches of 250 with *B. pahangi* mf as described above. After blood feeding, 25 mosquitoes were organized into small cardboard cartons in preparation for injection. Injections of dsRNA were carried out by delivering the dose via a glass microcapillary needle to the cervical membrane at the junction between the head and thorax, avoiding puncture of any sclerotized cuticle, and with the following modifications of an established protocol [64]. Prior to injection, infected mosquitoes were starved by removing sucrose pads 8 DPI. At 9 DPI, infected mosquitoes were injected with 250 µL of 1 µg/µL dsRNA, coinciding with the L2-to-L3 molt in the thoracic musculature [8]. Mosquitoes were injected in cohorts of 25, cohorts were immediately returned to 26˚C, and sucrose pads were replaced. Dead mosquitoes were removed daily until time of assay at 14 DPI, at which point mosquitoes were dissected to extract L3s for use in chemotaxis assays.

## Long-read sequencing in *B. malayi* adult males and females

Total RNA from *B. malayi* adult males and females was obtained from the FR3. RNA quality was assessed by a 2100 Bioanalyzer, converted to single-stranded cDNA and amplified using the SMARTer PCR cDNA Synthesis Kit (Takara Bio, Kusatsu, Japan), and Iso-Seq libraries were constructed with equimolar cDNA fractions (0.5× and 1×) with the SMRTbell Template Prep Kit 1.0 (Pacific Biosciences, Menlo Park, CA). Library quantity and quality were assessed by Qubit HS DNA (Thermo Fisher Scientific) and 2100 Bioanalyzer. Isoforms were clustered and polished from subreads with IsoSeq2, visualized with IGV [119], and annotated with BLAST [120].

## Cloning of *osm-9* and *tax-4* homologs

Primers directed toward the ATG start codon, stop codon, or 3′ UTR region of the Iso-Seq–generated gene model of *Bma-osm-9* and the predicted gene model of *Bma-tax-4* and *Bma-ocr-1/2a* (the *B. malayi* gene with the highest amino acid identity to *Cel-ocr-2*) were designed with Primer3 [118]. A full-length amplicon was produced with Phusion or Q5 polymerases (New England Biolabs). Amplicons were A-tailed with GoTaq Flexi (Promega, Madison, WI) and cloned into pGEM-T in JM109 competent cells (Promega). *C. elegans* N2 genomic DNA was extracted with the Qiagen DNeasy kit. An approximately 1.6-kb portion upstream of *Cel-osm-9* [32] and an approximately 3-kb portion upstream of *Cel-tax-4* [30] were amplified and cloned into pGEM-T as above. Final expression constructs were assembled with the HiFi Assembly kit (New England Biolabs) using amplicons generated with Q5 polymerase from the promoter and gene as two fragments and pPD95.75 (a gift from Andrew Fire [Addgene plasmid # 1494; http://n2t.net/addgene:1494; RRID: Addgene_1494]) double-digested with XbaI and EcoRI as the backbone. A BamHI restriction site was added between the promoter and open reading frames. *C. elegans osm-9* and *tax-4* open reading frames were amplified from plasmids (gifts from Shawn Xu [121] and Cornelia Bargmann [122], respectively) with Q5 polymerase. Each *C. elegans* gene was assembled into previously created expression vectors by replacing *B. malayi* genes with BamHI/EcoRI double digestions. The *unc-54* 3′ UTR of the pPD95.75 backbone was replaced in all constructs containing *osm-9* homologs by double-

digesting final constructs with EcoRI/BsiWI or PCR amplifying the entire plasmid without the *unc-54* 3′ UTR and assembling the resulting fragments with the *Cel-osm-9* 3′ UTR amplicon. Complete cloning protocols, including primer sequences and thermocycler programs, can be found in S4 File. All products were verified by Sanger sequencing. Expression vectors were injected into *C. elegans* hermaphrodites as described, and transgenic strains were used for rescue experiments.

### *C. elegans* sensory assays

Population chemotaxis assays were performed as described [123]. For each strain, five L4 worms from three independently derived lines were picked to each of five seeded NGM plates 5 days before the assay date. On assay day, worms from independent lines were washed off plates and pooled with M9 into a single tube per strain, washed with M9 three times, and washed once with water. For each of five 10-cm chemotaxis plates (2% agar, 5 mM $KH_2PO_4$/$K_2HPO_4$ [pH 6.0], 1 mM $CaCl_2$ and 1 mM $MgSO_4$) per strain, 1 μL of 1 M sodium azide was place on opposite sides of the plate and allowed to soak in with the plate lids removed. Once dry, 1 μL of cue and diluent were then placed at the same location as the sodium azide on opposing sides of the plate. Diacetyl (1:1,000; Santa Cruz Biotechnology, Santa Cruz, CA) and isoamyl alcohol (1:10; Thermo Fisher Scientific) in ethanol were used as cues for the *osm-9* and *tax-4* experiments, respectively. After the addition of cues, 100–200 worms were quickly pipetted to the center of assay plates. Excess water was removed with a Kimwipe (Kimberly-Clark, Irving, TX), and worms were gently spread with a platinum worm pick. Plates were left untouched on a benchtop for 60 minutes at room temperature (approximately 21°C), after which animals were counted in total and at each cue and control region. The CI of each plate was calculated as follows: $CI = (T − C) / (T + C + O)$, where T is the number of worms that were paralyzed at the test cue, C is the number at the control, and O is the number that had migrated to neither the test nor the control.

Benzaldehyde avoidance assays were performed as described [32,124]. Young adult animals were transferred to unseeded NGM plates, and 2 μL of benzaldehyde (Sigma-Aldrich) in a 20-μL borosilicate capillary was held in front of the animal's nose while the time to reversal was recorded. Five animals per strain were exposed a single time for each replicate, and a minimum of three replicates were performed.

Nose-touch reversal assays were performed as described [125]. Young adult animals were transferred to unseeded NGM plates and observed for reversal movement after colliding head-on with an eyelash. For each replicate, five animals per strain were observed for reversal after 10 successive collisions, and a minimum of three replicates were performed.

## Supporting information

**S1 Table. Spreadsheet of species included in the comparative analysis, clade designation, genome BioProject, and whether or not each species is represented in the trees.**
(XLSX)

**S1 Data. Chemoreceptor IQ-TREE consensus tree in Newick format.**
(TXT)

**S2 Data. TRP MrBayes consensus tree in Nexus format.** TRP, transient receptor potential.
(TXT)

**S3 Data. CNG MrBayes consensus tree in Nexus format.** CNG, cyclic nucleotide–gated.
(TXT)

**S4 Data. List of *C. elegans* chemoreceptor IDs. ID, identifier.**
(TXT)

**S1 Fig. Flow chart of comparative genomics pipeline as described in Materials and methods.**
(PDF)

**S2 Fig. Total chemoreceptor count as a function of genome contiguity.** The number of chemoreceptors in a given genome is not correlated to genome contiguity as measured by N50 (Spearman's rank-order correlation, $\rho = 0.182$, $p = 0.268$). Raw data can be found at https://github.com/zamanianlab/BrugiaChemo-ms.
(PDF)

**S3 Fig. Alternative plot of *B. malayi* head/tail RNA-seq (Fig 2).** Chemoreceptors are colored by superfamily annotation. Raw data can be found at https://github.com/zamanianlab/BrugiaChemo-ms. RNA-seq, RNA sequencing.
(PDF)

**S4 Fig. Motility analysis of L3 parasites after cooling and warming.** Worms move less after being cooled to room temperature, and motility subsequently increases after returning to 37˚C. Raw data can be found at https://github.com/zamanianlab/BrugiaChemo-ms. L3, third stage larvae.
(PDF)

**S5 Fig. Representative images of measured mosquito abdomens.** (A) Unfed. (B) Fed with unsupplemented blood. (C) Fed with blood supplemented with 5 mM NAM. (D) Fed with blood supplemented with 25 mM NAM. NAM, nicotinamide.
(PDF)

**S6 Fig. L3 recovery correlation plot.** The proportion of L3s recovered in the mosquito thorax does not correlate with the total L3s recovered per mosquito. Raw data can be found at https://github.com/zamanianlab/BrugiaChemo-ms. L3, third stage larvae.
(PDF)

**S7 Fig. *osm-9* nucleotide alignment.** The missing splice acceptor in the predicted gene model (Bm-osm-9_Bm1711.1) can be seen on the line starting with nucleotide 2,401.
(PDF)

**S8 Fig. *osm-9* amino acid alignment, including Inactive from *Drosophila melanogaster*.** The missing splice acceptor in the predicted gene model (Bm-osm-9_Bm1711.1), which led to a frameshift in the predicted amino acid sequence, can be seen on the line starting with amino acid 781.
(PDF)

**S9 Fig. *tax-4* amino acid alignment.** The mispredicted splice donor in the predicted gene model (Bm-tax-4_Bm7343.1), which led to a 7-aa deletion, can be seen on the line starting with amino acid 131.
(PDF)

**S10 Fig. qPCR results for knock-out and transgenic strains.** All transgenic strains had detectable RNA levels of the transgenes. Raw data can be found at https://github.com/zamanianlab/BrugiaChemo-ms. ND, not determined; qPCR, quantitative PCR.
(PDF)

**S11 Fig. Sensory assay data for *osm-9* strains with *unc-54* 3′ UTR.** Strains with the *unc-54* 3′ UTR were unable to rescue (A) defects in chemotaxis to diacetyl, (B) avoidance of concentrated benzaldehyde, or (C) reversal after light nose touch. Raw data can be found at https://github.com/zamanianlab/BrugiaChemo-ms.
(PDF)

**S12 Fig. Chemotaxis assay data for *osm-9* strains with *osm-9* 3′ UTR.** Strains with the *osm-9* 3′ UTR were unable to rescue defects in chemotaxis to diacetyl. Raw data can be found at https://github.com/zamanianlab/BrugiaChemo-ms.
(PDF)

**S1 File. List of all identified chemoreceptors with family and superfamily annotations.** Raw data can also be filtered and downloaded at https://zamanianlab.shinyapps.io/ChemoR/.
(CSV)

**S2 File. List of nematode species included in Fig 1D with assigned category and justification.**
(XLSX)

**S3 File. Template with instructions for assigning L3 coiling scores.** L3, third stage larvae.
(XLSX)

**S4 File. Complete protocols for all cloning efforts.**
(PDF)

## Acknowledgments

Some *C. elegans* strains were provided by the CGC. Parasite materials were provided by the NIH/NIAID Filariasis Research Reagent Resource Center (www.filariasiscenter.org). Sanger sequencing and RNA-seq were carried out at the University of Wisconsin-Madison Biotechnology Center. The authors would like to thank Tran To and Elena Garncarz for their assistance with the *C. elegans* behavioral assays, as well as members of the Zamanian laboratory for critical comments on the manuscript.

## Author Contributions

**Conceptualization:** Nicolas J. Wheeler, Mostafa Zamanian.

**Data curation:** Nicolas J. Wheeler, Zachary W. Heimark, Paul M. Airs, Mostafa Zamanian.

**Formal analysis:** Nicolas J. Wheeler, Zachary W. Heimark, Paul M. Airs, Alexis Mann, Mostafa Zamanian.

**Funding acquisition:** Lyric C. Bartholomay, Mostafa Zamanian.

**Investigation:** Nicolas J. Wheeler, Zachary W. Heimark, Paul M. Airs, Alexis Mann, Mostafa Zamanian.

**Methodology:** Nicolas J. Wheeler, Mostafa Zamanian.

**Project administration:** Lyric C. Bartholomay, Mostafa Zamanian.

**Resources:** Lyric C. Bartholomay, Mostafa Zamanian.

**Software:** Nicolas J. Wheeler, Mostafa Zamanian.

**Supervision:** Nicolas J. Wheeler, Lyric C. Bartholomay, Mostafa Zamanian.

**Validation:** Nicolas J. Wheeler, Mostafa Zamanian.

**Visualization:** Nicolas J. Wheeler, Paul M. Airs, Mostafa Zamanian.

**Writing – original draft:** Nicolas J. Wheeler, Lyric C. Bartholomay, Mostafa Zamanian.

**Writing – review & editing:** Nicolas J. Wheeler, Zachary W. Heimark, Paul M. Airs, Alexis Mann, Lyric C. Bartholomay, Mostafa Zamanian.

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
