## [Editor Report · Decision Letter 0]

22 Jul 2019

Dear Dr Zamanian, 

Thank you for submitting your manuscript entitled "Genetic and functional diversification of chemosensory pathway receptors in mosquito-borne filarial nematodes" for consideration as a Research Article by PLOS Biology.

Your manuscript has now been evaluated by the PLOS Biology editorial staff as well as by an academic editor with relevant expertise and I am writing to let you know that we would like to send your submission out for external peer review.

**Important**: Please also see below for further information regarding completing the MDAR reporting checklist. The checklist can be accessed here: https://plos.io/MDARChecklist

Please re-submit your manuscript and the checklist, within two working days, i.e. by Jul 24 2019 11:59PM.

Kind regards,

Lauren A Richardson, Ph.D

Senior Editor

PLOS Biology

INFORMATION REGARDING THE REPORTING CHECKLIST:

PLOS Biology is pleased to support the "minimum reporting standards in the life sciences" initiative (https://osf.io/preprints/metaarxiv/9sm4x/). This effort brings together a number of leading journals and reproducibility experts to develop minimum expectations for reporting information about Materials (including data and code), Design, Analysis and Reporting (MDAR) in published papers. We believe broad alignment on these standards will be to the benefit of authors, reviewers, journals and the wider research community and will help drive better practise in publishing reproducible research. 

We are therefore participating in a community pilot involving a small number of life science journals to test the MDAR checklist. The checklist is intended to help authors, reviewers and editors adopt and implement the minimum reporting framework. 

IMPORTANT: We have chosen your manuscript to participate in this trial. The relevant documents can be located here:

MDAR reporting checklist (to be filled in by you): https://plos.io/MDARChecklist

**We strongly encourage you to complete the MDAR reporting checklist and return it to us with your full submission, as described above. We would also be very grateful if you could complete this author survey:

https://forms.gle/seEgCrDtM6GLKFGQA

Additional background information:

Interpreting the MDAR Framework: https://plos.io/MDARFramework

Please note that your completed checklist and survey will be shared with the minimum reporting standards working group. However, the working group will not be provided with access to the manuscript or any other confidential information including author identities, manuscript titles or abstracts. Feedback from this process will be used to consider next steps, which might include revisions to the content of the checklist. Data and materials from this initial trial will be publicly shared in September 2019. Data will only be provided in aggregate form and will not be parsed by individual article or by journal, so as to respect the confidentiality of responses. 

Please treat the checklist and elaboration as confidential as public release is planned for September 2019.

We would be grateful for any feedback you may have.

---

## [Decision Letter · Decision Letter 1]

9 Aug 2019

Dear Dr Zamanian,

Thank you very much for submitting your manuscript "Genetic and functional diversification of chemosensory pathway receptors in mosquito-borne filarial nematodes" for consideration as a Research Article at PLOS Biology. Your manuscript has been evaluated by the PLOS Biology editors, an Academic Editor with relevant expertise, and by several independent reviewers.

As you will read, the reviewers appreciated the potential impact of your work. However, they do raise some concerns that will need to be addressed in a revision. In particular, they request additional controls to validate the knockdown experiments and to show that the channels are properly expressed and localized.

In light of the reviews (below), we will not be able to accept the current version of the manuscript, but we would welcome resubmission of a much-revised version that takes into account the reviewers' comments. We cannot make any decision about publication until we have seen the revised manuscript and your response to the reviewers' comments. Your revised manuscript is also likely to be sent for further evaluation by the reviewers.

Your revisions should address the specific points made by each reviewer. Please submit a file detailing your responses to the editorial requests and a point-by-point response to all of the reviewers' comments that indicates the changes you have made to the manuscript. In addition to a clean copy of the manuscript, please upload a 'track-changes' version of your manuscript that specifies the edits made. This should be uploaded as a "Related" file type. You should also cite any additional relevant literature that has been published since the original submission and mention any additional citations in your response. 

Before you revise your manuscript, please review the following PLOS policy and formatting requirements checklist PDF: http://journals.plos.org/plosbiology/s/file?id=9411/plos-biology-formatting-checklist.pdf. It is helpful if you format your revision according to our requirements - should your paper subsequently be accepted, this will save time at the acceptance stage.

Please note that as a condition of publication PLOS' data policy (http://journals.plos.org/plosbiology/s/data-availability) requires that you make available all data used to draw the conclusions arrived at in your manuscript. If you have not already done so, you must include any data used in your manuscript either in appropriate repositories, within the body of the manuscript, or as supporting information (N.B. this includes any numerical values that were used to generate graphs, histograms etc.). For an example see here: http://www.plosbiology.org/article/info%3Adoi%2F10.1371%2Fjournal.pbio.1001908#s5.

For manuscripts submitted on or after 1st July 2019, we require the original, uncropped and minimally adjusted images supporting all blot and gel results reported in an article's figures or Supporting Information files. We will require these files before a manuscript can be accepted so please prepare them now, if you have not already uploaded them. Please carefully read our guidelines for how to prepare and upload this data: https://journals.plos.org/plosbiology/s/figures#loc-blot-and-gel-reporting-requirements.

Upon resubmission, the editors will assess your revision and if the editors and Academic Editor feel that the revised manuscript remains appropriate for the journal, we will send the manuscript for re-review. We aim to consult the same Academic Editor and reviewers for revised manuscripts but may consult others if needed.

We expect to receive your revised manuscript within two months. Please email us (plosbiology@plos.org) to discuss this if you have any questions or concerns, or would like to request an extension. At this stage, your manuscript remains formally under active consideration at our journal; please notify us by email if you do not wish to submit a revision and instead wish to pursue publication elsewhere, so that we may end consideration of the manuscript at PLOS Biology.

When you are ready to submit a revised version of your manuscript, please go to https://www.editorialmanager.com/pbiology/ and log in as an Author. Click the link labelled 'Submissions Needing Revision' where you will find your submission record. 

Sincerely,

Lauren A Richardson, Ph.D

Senior Editor

PLOS Biology

Reviews

Reviewer #1: 

These authors report their findings studying chemosensory signaling in filarial nematodes; an important yet vastly understudied area. Using a variety of techniques they perform a broad study of chemoreceptive behaviors in filarial nematodes. They find that the chemoreceptor family size is correlated with free-living life stages and they describe the complement of putative chemoreceptors in filarial nematodes. They examine the expression of putative chemoreceptors in Brugia using different stages of the nematodes, and they also examined this expression in different regions of the nematodes. They then begin to explore mechanistic level detail of chemoreception in these nematodes, with a focus on the TRP channel OSM-9 and the CNG channel subunit TAX-4. Most of the claims made are novel and are well supported. Overall the experiments performed are elegant and have proper controls, and I appreciated the care and restraint used by the authors in interpreting their data. There are several findings that I thought were especially exciting and I look forward to seeing these ideas further developed in future studies. The paper contains a large amount of work, but I have suggested the inclusion of an additional control experiment that I think would strengthen their findings regarding the role of osm-9 and tax-4 in chemotaxis. I think the paper is outstanding in its discipline because: 1) the pan-phylum level of chemoreceptor analysis is the largest and most inclusive yet reported, and 2) the multi-tiered approach employing comparative genomics, transcriptomics, reverse genetics, and pharmacology to studying the behavior of filarial nematodes is powerful and unlike anything else published in this field. It sets a new standard for research in filarial nematodes and provides some important tools for enhancing future research. The findings are of broad interest and would serve to shape future studies on host-seeking behavior among parasitic helminths and the specific role of CNG and TRP channels in these behaviors. I do have a few concerns and suggestions, but overall this is very nice work addressing an important and understudied field. 

Major Issues:

1) The authors performed a great intra-mosquito RNAi experiment. I appreciated the disclosure that while they attempted to confirm knockdown by qPCR they were unable to measure expression for reasons they provided. While a nonspecific RNAi control was good to include, I think the authors should include knockdown of a chemoreceptor not thought to be involved in this process, as is a common practice in C. elegans research. This would illustrate that the behavior is specific to the knockdown of osm-9 and tax-4 rather than successful knockdown of a channel receptor in the microfilariae. 

I appreciated that the authors included a nice control when examining the effect of treating with nicotinamide by measuring the relative expression of Bp-osm-9 in L3s under different conditions up to 4 hours after being removed from the mosquito host. In the subsequent paragraph of the results the authors discuss the loss of chemotactic response of nematodes that were maintained overnight under standard culture conditions. I also appreciated their interpretation of the loss of chemotaxis toward serum after an overnight culture.

Minor issues

There are some minor issues scattered throughout the manuscript that need to be fixed.

1) The figures should be reassessed for correct numbering. The results section on “Treatment with a nematode TRPV agonist inhibits chemoattraction…” erroneously refers to incorrect panels of figure 4. The current reference to Figure 4A should refer to Figure 4B. The current reference to Figure 4B should probably be moved to the previous sentence and a new reference to Figure 4C should be put in its place. The current reference to Figure 4C should be changed to figure 4D. The current references to Figure 4D should be switched to reference Figure 4E.

There are also some modifications that could be made to the figures to facilitate each of interpretation. One suggestion is that the authors could include a gene name title for Figure 3 panels C-F (e.g. osm-9 for panel C) that would allow readers to immediately identify the gene trees without having to bounce between the legend and the figure. The little diagram regarding the chemotaxis assay setup used in figure 4 and figure 7 should be explained in the legend. I realize that it is explained in the methods section, but it should also be explained in the legends of these figures.

2) The references need to be evaluated and corrected. For example, there are many references with “[internet]” in them. For example, references 5, 18, 21, 34, 46, 49, 76, and 77 have this error. There are also several places in the manuscript where instead of using numbers the last names of authors are used. This occurs in several places including in the 2nd paragraph of the introduction and at the end of the 4th paragraph of the introduction, and again in the last paragraph of the discussion. PLoS formatting style requires that numbers be used throughout the text rather than author names.

3) In the 2nd page of the results section the authors discuss how their pan-phylum analysis revealed a significant correlation between chemoreceptor gene count and the presence and nature of free-living or environmental stages of each nematode species life cycle. While the analysis in this paper includes more genomes than previous studies, and is more thorough and quantitative in their analysis, other researchers have previously reported this finding and have hypothesized that species with fewer environmental stages have a more compact repertoire of chemoreceptors. That the current study supports this previously reported hypothesis is interesting, and these authors should cite the previous studies [1, 2].

References:

1. Srinivasan, J., Dillman, A.R., Macchietto, M.G., Heikkinen, L., Lakso, M., Fracchia, K.M., Antoshechkin, I., Mortazavi, A., Wong, G., and Sternberg, P.W. (2013). The draft genome and transcriptome of Panagrellus redivivus are shaped by the harsh demands of a free-living lifestyle. Genetics 193, 1279-1295.

2. Thomas, J.H., and Robertson, H.M. (2008). The Caenorhabditis chemoreceptor gene families. BMC biology 6, 42.

Reviewer #2: 

This is a well-written manuscript that opens promising new avenues in research on the the basic biology of host-parasite interactions. It is highly innovative and is based on an intriguing set of hypotheses that are pertinent to answering questions about how filarial parasites detect and respond to chemical cues as they develop in two different hosts. It certainly is worth publishing in PLoS-Biology.

I have only minor concerns. Most importantly, the failure to detect functional expression of the two Brugia proteins in complementation experiments in C. elegans is not really conclusive. The analysis would be strengthen by the inclusion of the respective positive control (the C. elegans gene) to demonstrate success of the technique in the authors laboratory.

More minor concerns: 

1. The text is sprinkled with references cited by name instead of number, an oversight that needs to be cleaned up (renumbering may be necessary). Please also ensure appropriate italicization in the journal titles.

2. Reference 18 seems to incomplete

3. Page 14: How do you know that NAM was restricted to the midgut and did not distribute throughout the mosquito body?

4. Page 19: How many 1 mm segments were pooled for these analyses? Were independent biological replicates analyzed?

Reviewer #3: 

Review of manuscript #PBIOLOGY-19-02073R1 by Wheeler, Zamanian, et al. entitled “Genetic and functional diversification of chemosensory pathway receptors in mosquito-borne filarial nematodes”. 

The authors report a thorough phylogenetic study of chemoreceptors and downstream signaling elements in Brugia malayi. Considering the need for novel interventions in human filariasis, their finding of a reduced and highly diverged repertoire of chemoreceptors compared to free-living nematodes represents a significant contribution to the literature on lymphatic filariasis and is consistent with the life history of this lymphatic dwelling filaria that inhabits either a mammalian or a mosquito host with only a brief transitional exposure to the external environment during the transition between emergence from the labial sheath of the mouthparts and invasion of the mosquito bite wound. Notably, the authors have discovered changes in expression patterns of B malayi chemoreceptors that coincide with key migratory events and transitions between host and vector body compartments. Intriguingly, exposure to nicotinamide, an agonist of the sensory neuronal TRP channel OSM-9, suppresses outward migration of microfilariae from the midgut, suggesting that OSM-9-dependent chemotaxis is required for this vital migratory transition by the parasite. This is a testable hypothesis that could form the basis for future studies. Significantly, they also show that worms exposed to dsRNAs with sequence homology to the TRP channel encoding gene Bm-osm-9 and the CNG channel encoding gene Bm-tax-4 are impaired in their ability to execute diacetyl- and isoamyl alcohol-stimulated chemotaxis. 

This is a well written paper with many positive aspects. The phylogenetic study of B. malayi chemoreceptors and downstream channels involved in signaling is very well done insofar as I can judge it. Also, the chemotactic and behavioral assays are careful and well thought out. In particular, the conduct of the coiling/uncoiling assay, which could be somewhat subjective in its interpretation, was done in blinded fashion to eliminate bias. Also, studies on the anatomical distribution of B. malayi larvae within susceptible Aedes aegypti were well designed and revealed a decrement in thoracic muscle burdens of L3 resulting from nicotinamide exposure, which supports a hypothesis that OSM-9 function is essential for outward migration of microfilariae from the vector midgut. 

There are however instances (the RNAi study and the heterologous complementation studies) in which some rather important control data were not or could not be gathered, and these detract from an otherwise excellent study. These are listed as substantive issues below. 

As stated, the paper is very well written and was generally an easy read. I only noted a few minor editing issues for the authors’ attention.

SUBSTANTIVE ISSUES: (Note: I believe that addressing substantive points 2 and 3 is essential to support the current conclusions)

1. Pg. 1, line 11 of main text: It might be worth noting here that there are also constraints on the use of ivermectin in onchocerciasis patients where the risk of Onchocerca volvulus and Loa loa co-infection is present.

2. I can absolutely understand the difficulty in confirming knockdown of Bm-osm-9- and Bm-tax-4-specific messages within the experimental context of the RNAi studies, given the small numbers of L3 present and the imperative to recover sufficient numbers of them for chemotaxis assay. However, I still think it is crucial to provide data, even data gathered outside of the experimental context, that confirms knockdown efficiency and specificity of the protocol used. As a second-best approach, couldn’t some “in squito” dsRNA exposures, without the constraint of gathering L3 for chemotaxis assay, be dedicated to gathering this important control data for each target gene and the control lacZ?

3. Similarly, the negative findings of the heterologous complementation studies would be bolstered by inclusion of a positive control involving C. elegans osm-9(ky10) and Tax4(p768) mutants transformed with homologous (C. elegans-derived) coding sequences under their respective promoters. Given the negative results of the heterologous rescue attempts, it seems essential to prove that rescue with homologous sequences is possible before concluding divergence of function of the parasite elements. 

4. The paper would be improved by a more detailed account in Materials and Methods of how the transgenic lines used in the heterologous complementation studies were derived. For example, were several independently derived lines of C. elegans transformed with the parasite sequences assessed for rescuing ability? Given the stochastic process of forming episomal transgene arrays in C. elegans, it is generally considered essential to base conclusions on studies of at least three independently derived lines. This shouldn’t be burdensome as independent lines can be derived from hermaphrodites that are singled after microinjection with their progeny grown in isolation. 

MINOR EDITING ISSUES:

Note: Numbered lines and pages would greatly facilitate review of the revised ms. 

5. Pg. 4 of main text, line 26: suggest deleting “worms”. It’s redundant.

6. Pg. 4, line 31: suggest “repelled” instead of “repulsed”. 

7. Pg. 4, line 36: suggest “…specific defect in chemotaxis…”. 

8. Pg. 4, line 37: suggest inserting “that” after “ensure”.

9. Pg. 5, lines 1-3: suggest moving the reference “(Figure 4D)” in line 3 to come after “extraction” in line 1 and adding a reference to Figure 4E after “serum-free media” in line 3. 

10. Pg. 8, line 18 and elsewhere in the paper: the word “data” is plural, so make it “Our data suggest”. Same in line 25 and several other places (eg. caption to Fig. 11).

Reviewer #4: 

In this study, Wheeler et al. provide some of the first insights into the mechanisms of chemosensation in filarial nematodes. While there is now extensive evidence that mammalian-parasitic nematodes with an environmental infective stage actively chemotax to host and environmental cues, much less is known about the chemosensory behaviors of vector-transmitted parasitic nematodes. This study identifies putative chemoreceptors in Brugia (and also provides a much more thorough analysis of putative chemoreceptors across nematode species than was provided in previous studies), and shows that their expression varies across life stages. The authors then implicate Brugia osm-9 and tax-4 is larval chemotaxis, providing the first insights into chemosensory signaling pathways in these parasites. Together, these results represent an important first step in understanding chemosensation in filarial parasites. In addition, the detailed chemoreceptor annotations in this paper will be a valuable resource for those interested in nematode chemosensation in general.

Major comments:

1. In Figure 1c, I’m not clear on how it was determined which species have more environmental exposure for some of the categories. For example, among the EPNs, skin-penetrating nematodes, and passively ingested nematodes such as H. contortus, why are some considered to have more environmental exposure than others, when these worms have a similar third-larval stage in the environment. A bit more justification is needed for how some of the worms were ordered within each subcategory.

2. Are all the parasite genomes equally complete? Could some of the differences in numbers of predicted receptors be due to the quality of the available genomes?

3. In Figure 2a and 2b, the colors that indicate the superfamilies are a bit confusing. Is it possible to put dividing lines between the genes, so that it’s clear how many genes are represented by each colored region?

4. For the RNASeq analysis shown in Figure 2b, did the authors also perform RNASeq analysis on the middle region of the worm, where chemoreceptor expression is not expected (or should be lower)? It would be nice to have this data as a control.

5. Figure 4: What is the scale for the chemotaxis assays? It looks like the worms would hardly need to move to get to the C or T regions. How motile are the worms? For the motility assay, can the authors provide data on the extent to which the worms disperse across the plate? They can clearly move out of the center region, but it’s not clear how far they are capable of moving.

6. Regarding Figure 5:

a) Is the coiling response reversable, for example if the worm is put back at 37C? If it’s not reversible, isn’t it possible the response results from muscle or tissue damage, or some other more general response?

b) Does adding NAM to L3s that have adapted to RT cause them to become more active by mimicking the effect of heat?

c) I don’t understand the description of sample sizes. Did the authors test only 3 worms per condition? Or does each biological and technical replicate involve multiple worms? If only 3 worms were used, this seems very low. Did the authors perform power analysis to get a sense of how many worms they should be testing? 

d) The coiling score seems like it could be made more rigorous. First, it’s not explicitly stated how the coiling score is being determined. What are the criteria for rating worms on a scale of 0 to 5? The descriptions in the methods section is really vague. Also, the data would be more convincing if a more quantitative measure were used. If there are videos of the worms anyway, isn’t it straightforward to measure curvature quantitatively?

7. Figure 8: I’m not sure what can be concluded from this experiment. Is there any evidence the channels are actually being expressed in C. elegans? It doesn’t look like they were expressed using an SL2::GFP, making it difficult to determine whether the channels are non-functional in C. elegans or are not being expressed sufficiently. Given that it’s hard to conclude much from this data, perhaps this figure can be moved to supplemental?

Minor comments:

1. Starting off with a diagram of the life cycle of Brugia would be helpful for the non-expert.

2. Bottom of p. 9: Should this say, “These pathways have likely evolved to reflect the diversity of nematode life-history traits and environmental cues encountered by different *nematode* species” (i.e., “nematode” instead of “parasite”)?

3. The expression data in Figure 2a is difficult to interpret. Is there a legend for the purple shading? Also, it’s hard to see the purple shading when it’s inside the circle, and it’s confusing the way Figure 2b is discussed before this part of Figure 2a in the main text. It might be nice to separate out the life-stage expression data into a separate figure.

4. Is there expression data for the signaling proteins (tax-2, tax-4, osm-9, etc.) showing whether they are expressed in the head region (like what is shown in Figure 2b for the chemoreceptors)? This would provide some additional information about which of these genes is likely to have a chemosensory function in filarial nematodes.

5. For Figure 3c-f, it would be helpful to indicate at the top of each figure which gene is shown.

6. Figure 4: How many worms were typically used for each chemotaxis assay? Also, in Figure 4b, why is the mean given only for the untreated condition and not the NAM condition? For Figure 4e, are there untreated controls with freshly extracted parasites that were performed in parallel? It would be nice to have data for freshly extracted parasites and 1 DPE parasites that were tested in parallel shown side-by-side.

7. Figure 6a: Is there a reason not to label every point along the x-axis?

8. Throughout the manuscript, it’s not clear exactly how many worms were tested for each assay. Also, are the biological replicates different groups of mosquitoes and the technical replicates different trials? Were the different trials performed on the same day or separate days? How many mosquitoes were used for each biological replicate? Are the data points shown the means from each biological replicate?

---

## [Decision Letter · Decision Letter 2]

4 Mar 2020

Dear Dr Zamanian,

Thank you very much for submitting a revised version of your manuscript "Genetic and functional diversification of chemosensory pathway receptors in mosquito-borne filarial nematodes" for consideration as a Research Article at PLOS Biology. This revised version of your manuscript has been evaluated by the PLOS Biology editors and by the original Academic Editor and reviewers 2, 3, and 4.

In light of the reviews (below), we are positive about your manuscript and are thus pleased to offer you the opportunity to address the remaining points from the reviewers in a revised version that we anticipate should not take you very long. We will then assess your revised manuscript and your response to the reviewers' comments and we may consult the reviewers again.

We expect to receive your revised manuscript within 1 month.

**IMPORTANT - SUBMITTING YOUR REVISION**

Your revisions should address the specific points made by each reviewer. Having discussed these comments with the Academic Editor, we think you should address the points raised either experimentally or, if these are too challenging, textually. In other words, the inconclusive rescue results should be moved to supplemental and extreme care should be taken in their interpretation in the text.

Please submit the following files along with your revised manuscript:

*Resubmission Checklist*

*Published Peer Review*

*PLOS Data Policy*

*Blot and Gel Data Policy*

Sincerely,

Gabriel Gasque, Ph.D., 

Senior Editor

PLOS Biology

REVIEWS:

Reviewer's Responses to Questions

Reviewer #2: The authors have responded constructively and positively to my concerns. I think this is an important contribution to the literature and recommend that it now be processed for publication.

Reviewer #3: Review of manuscript #PBIOLOGY-19-02073R2 by Wheeler, Zamanian, et al. entitled "Genetic and functional diversification of chemosensory pathway receptors in mosquito-borne filarial nematodes". 

This second revision is significantly improved and remains a thorough phylogenetic study of chemoreceptors and downstream signaling elements in Brugia malayi. Considering the need for novel interventions in human filariasis, their finding of a reduced and highly diverged repertoire of chemoreceptors compared to free-living nematodes represents a significant contribution to the literature on lymphatic filariasis and is consistent with the life history of this lymphatic dwelling filaria that inhabits either a mammalian or a mosquito host with only a brief transitional exposure to the external environment during the transition between emergence from the labial sheath of the mouthparts and invasion of the mosquito bite wound. Notably, the authors have discovered changes in expression patterns of B malayi chemoreceptors that coincide with key migratory events and transitions between host and vector body compartments. Intriguingly, exposure to nicotinamide, an agonist of the sensory neuronal TRP channel OSM-9, suppresses outward migration of microfilariae from the midgut, suggesting that OSM-9-dependent chemotaxis is required for this vital migratory transition by the parasite. This is a testable hypothesis that could form the basis for future studies. Significantly, they also show that worms exposed to dsRNAs with sequence homology to the TRP channel encoding gene Bm-osm-9 and the CNG channel encoding gene Bm-tax-4 are impaired in their ability to execute diacetyl- and isoamyl alcohol-stimulated chemotaxis. 

In this second revision, I believe that the authors have satisfactorily addressed most of the substantive issues that I noted in my review. In reading the rejoinder to other reviewers' comments it appears that they have similarly addressed substantive issues to the extent that this very challenging filarial system allows. I have only one substantive and one minor editing point to list with regard to this second revision. 

SUBSTANTIVE ISSUE

The issue I raised about confirming efficient knock-down of target transcripts remains, as the authors were unable to collect sufficient parasite material either in the context of their experiments involving Bm-osm-9- and Bm-tax-4-specific knockdown or in the context of additional supporting experiments to accurately confirm suppression of target transcripts. I can accept that this technical hurdle is insurmountable at this time but urge the authors to consider devising approaches to addressing it in future projects and publications. 

MINOR EDITIORIAL ISSUE

Strictly speaking the word "data" is plural, so issues with subject-verb agreement arising from its use as a singular are commonplace. The authors carefully addressed points in the main text relating to this. However, a few lapses are still seen in the figure captions in the second revision. Captions to Figures 4, 6, 7 and 8 all contain some variation on "Data….represents". It should be "Data…..represent". It might be a good idea to check any captions to supplemental figures along this line. 

Reviewer #4: Overall, the authors have done a thorough job in addressing the reviewers' comments and the revised manuscript is much improved. I expect this paper will soon be a landmark in the field. While it would have been nice to see RNAi for another endogenous gene as a control to increase confidence in the RNAi data, I understand that these experiments are not feasible at this time. I have only a few remaining comments which I think should be addressed prior to publication.

Major comments:

1. In my opinion, the biggest issue remaining with the revised paper is with the rescue data shown in Figures 8 and 9.

First, I would suggest removing some of the inconclusive rescue data from the paper. Since using the unc-54 3' UTR didn't work, I don't think anything useful is gained by including this data. Also, for the diacetyl experiment, you still can't conclude anything since the C. elegans rescue didn't work. I don't think having this data in the paper is informative, and if it is included, I might at least move it to a supplemental figure.

Second, it looks like the B. malayi cDNA was used directly for the rescue experiment. However, expression of cDNA from other nematodes in C. elegans often requires the placement of internal syntrons in the cDNA. A syntron before the gene is sufficient for C. elegans cDNA expression, but often not for the expression of cDNA from distantly related nematodes. I would strongly suggest repeating this experiment using synthesized cDNA with internal introns, or at least using this approach going forward. From the data shown, I don't think anything can be concluded other than that partial rescue is possible in some cases. 

Third, if the C. elegans osm-9 promoter fragment being used is the issue with the diacetyl experiment, there are AWA-specific promoters that could be used instead.

If it's not possible to repeat these experiments at this point - I do realize these experiments are extremely difficult and time-consuming - I would at least be very careful about concluding anything beyond the fact that partial rescue occurred. In particular, I would remove the sentence about how partial rescue may suggest subfunctionalization, or at least qualify that this could be due to issues with transgene expression.

2. The methods section states that for the chemotaxis assays, assays were counted if C + T > 2. This seems very low. I doubt the CI is very meaningful when only two worms are involved in the calculation. I would suggest investigating this possible issue by repeating the calculations using only assays where C + T > 5 to see if the data look similar.

Minor comments:

1. In the legend for Figure 1, panels B and C appear to be switched.

2. Line 137: Change to, "that likely aid in copulation."

3. Fig. 2: Can the authors speculate as to why there are so many chemoreceptors expressed in embryos?

4. Figure 4E: axis label says "Indexi" instead of "Index."

5. Figure 5B: The text (lines 237-238) doesn't seem consistent with the figure and figure legend. The text makes it sound like the images in 5B show wild-type phenotypes at different temperatures, whereas the figure looks like it's showing control vs. NAM-treated.

6. Figure 5: Why were one-sided t-tests used instead of two-sided t-tests?

7. Figure 6B and F: Why were t-tests used instead of ANOVAs? It looks like 3 groups are being compared.

8. If there's any way to get Figure 7 to fit in the normal orientation, that would be preferable.

9. Is there a reason not to use violin plots in Figure 7? I think the violin plots used elsewhere in the manuscript are a really nice way of representing the data.

10. Line 295-296: "To our knowledge, this is the first time that tax-4 or osm-8 have been shown to have a specific function in chemotaxis in a parasitic nematode." This is not quite correct. A recent paper showed a role for tax-4 in plant-parasitic nematodes using an RNAi approach:

Nagendrappa, S.K.T., Dutta, T.K., Chaudhary, S., von Reuss, S.H., Williamson, V., and Rao, U. (2019) Homologs of Caenorhabditis elegans chemosensory genes have roles in behaviour and chemotaxis in the root-knot nematode Meloidogyne incognita. Mol Plant Microbe Interact, e-pub ahead of print.

11. Line 305 needs proof-reading.

12. Line 330: change "addition" to "additional"

---

## [Editor Report · Decision Letter 3]

27 Mar 2020

Dear Dr Zamanian,

Thank you for submitting your revised Research Article entitled "Genetic and functional diversification of chemosensory pathway receptors in mosquito-borne filarial nematodes" for publication in PLOS Biology. I have now discussed your new version with the Academic Editor, and we're delighted to let you know that we're editorially satisfied with your manuscript. 

However before we can formally accept your paper and consider it "in press", we also need to ensure that your article conforms to our guidelines. A member of our team will be in touch shortly with a set of requests. As we can't proceed until these requirements are met, your swift response will help prevent delays to publication. Please also make sure to address the data requests noted at the end of this email.

*Copyediting*

*Published Peer Review History*

*Early Version*

*Submitting Your Revision*

Sincerely,

Gabriel Gasque, Ph.D., 

Senior Editor

PLOS Biology

DATA POLICY:

We note that you have stated, in your Data Availability Statement, that ““All comparative genomics, phylogenetics, data analysis, and data visualization pipelines are publicly available at https://github.com/zamanianlab/BrugiaChemo-ms. Short-read and long-read sequencing data has been deposited into NIH BioProjects PRJNA548881 and PRJNA548902, respectively. All other data are contained with the paper and/or Supporting Information files.”

However, we also ask that you provide all individual quantitative observations that underlie the data summarized in the figures and results of your paper. These data can be made available in one of the following forms:

Regardless of the method selected, please ensure that you provide the individual numerical values that underlie the summary data displayed in the following figure panels: Fig. 4 B, C, D and E, Fig. 5 C and D, Fig. 6 A, B, C, D, E and F, Fig. 7 A, B and C, Fig. 8 A, B and C, Fig. 9, Fig. S2, Fig. S3, Fig. S4, Fig. S6 and Fig. S10.

Please also ensure that the figure legends in your manuscript include information on where the underlying data can be found, and ensure your supplemental data file/s has a legend.

For manuscripts submitted on or after 1st July 2019, we require the original, uncropped and minimally adjusted images supporting all blot and gel results reported in an article's figures or Supporting Information files. We will require these files before a manuscript can be accepted so please prepare and upload them now. Please carefully read our guidelines for how to prepare and upload this data: https://journals.plos.org/plosbiology/s/figures#loc-blot-and-gel-reporting-requirements.

---

## [Editor Report · Decision Letter 4]

20 May 2020

Dear Dr Zamanian,

On behalf of my colleagues and the Academic Editor, Piali Sengupta, I am pleased to inform you that we will be delighted to publish your Research Article in PLOS Biology. 

Early Version

PRESS 

Kind regards,

Vita Usova

Publication Assistant, 

PLOS Biology

on behalf of

Gabriel Gasque,

Senior Editor

PLOS Biology